# A Causal Theoretical Framework for Open Set Domain Adaptation

## Abstract

Open Set Domain Adaptation (OSDA) faces two critical challenges: the emergence of unknown classes in the target domain and changes in observed distributions across domains. Although numerous studies have proposed advanced algorithms, recent experimental results demonstrate that the classical Empirical Risk Minimization (ERM) approach still delivers state-of-the-art performance. However, few theories can effectively explain this disputed phenomenon. To address the theoretical gap, we focus on constructing a causal theoretical framework for OSDA. We formulate the novel concepts of the Fully Informative Causal Invariance Model (FICIM) and the Partially Informative Causal Invariance Model (PICIM). Subsequently, We derive an OSDA theoretical bound to prove that the ERM performs well when the source domain follows FICIM, while it performs poorly when the source domain follows PICIM. The different results may be attributed to the varying amounts of available information when bounding the target domain's stable expected risk. Finally, across different datasets, we conduct extensive experiments on the FICIM and PICIM source domains to validate the effectiveness of our theoretical results.

## 1 Introduction

Open Set Domain Adaptation (OSDA) represents a realistic challenge in domain adaptation (Fang et al., 2020). There is a great need to solve OSDA in the real world. For instance, autonomous driving AI is often trained in simulated environments but must operate in complex real-world scenarios that may involve unseen targets (Li et al., 2023; Oza et al., 2023). Chatbots can become more intelligent via detecting unknown expressions and prompting users to explain them (Abdaljaleel et al., 2024). Furthermore, if AI overlooks unknown instances, it may become overly confident, resulting in serious hallucinations and safety issues (Xu et al., 2023; Zhu et al., 2024).

OSDA is more challenging than other domain adaptation problems, as illustrated in Fig. 1. The first challenge is that unknown classes appear in the target domain. The second challenge is the observed distributions of data which changes across domains. Existing domain adaptation studies rely on strong assumptions on observed distributions of inputs and labels. One key assumption is the covariate shift assumption $p_S(\mathbf{x}) \neq p_T(\mathbf{x})$ while $p_S(\mathbf{y}|\mathbf{x}) = p_T(\mathbf{y}|\mathbf{x})$ (Pan et al., 2010), which states that the conditional distribution of the labels (given the input x) is invariant across domains. However, such assumptions are too restrictive for high-dimensional data due to dimension redundancy and the lack of direct causal relationships or correlations between the original high-dimensional data and the prediction task (Niyogi, 2013; Bengio et al., 2013; Wang et al., 2021; 2024). Although most of the existing literature has

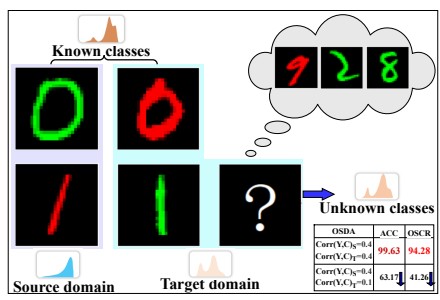

Figure 1: In this OSDA scenario, 1) the unknown classes appear in the target domain, 2) the digit color is a latent attribute correlated with the image $X$ and digit label $Y$, 3) the digit color is positively correlated with the label in the source domain and is negatively correlated with the label in the target domain, 4) $p_S(\mathbf{y}|\mathbf{x}) \neq p_T(\mathbf{y}|\mathbf{x})$ due to the correlation between color and label.

claimed improved performance of OSDA using different algorithms (Fang et al., 2020; Chen et al., 2021; Zhou et al., 2021; Qu et al., 2024; Yang et al., 2024), the performance gains have been reported to be overestimated, with the classic Empirical Risk Minimization (ERM) method remaining state-of-the-art (Vaze et al., 2022; Wu et al., 2023; Vaze et al., 2024; Qu et al., 2024). This performance controversy motivates us to develop a theoretical risk decomposition.

To address the above issues, we propose a theoretical framework based on the invariant causal mechanisms (Fan et al., 2023; Yuan et al., 2024; Yao et al., 2024; Zhang et al., 2024) from causal theory to understand how stable causal mechanisms[1] facilitate knowledge transfer and explain why algorithms like ERM succeed in some scenarios while failing in others. This framework includes two models: the Fully Informative Causal Invariance Model (FICIM) and the Partially Informative Causal Invariance Model (PICIM). Via distinguishing FICIM and PICIM, we can better define the conditions under which domain adaptation methods are effective and derive bounds on the expected risk in the target domain.

More technically, we define the stable expected risk with invariant connections across domains and derive a theoretical bound of the stable expected risk for OSDA. Furthermore, our bound explains which risk minimization strategies should be employed under which conditions. Our theory addresses the theoretical performance controversy between ERM and other methods: 1) The ERM of a source domain following FICIM can provide sufficient information to bound the stable expected risk of the target domain; 2) The stable expected risk of the target domain cannot be bounded by the ERM of a source domain following PICIM. In addition, generating source domain data that adheres to FICIM is beneficial for model training or fine-tuning, especially for large language models (LLMs). We conduct extensive experiments on multiple FICIM and PICIM datasets to validate the reliability of our theoretical results.

The significant contributions of this work are summarized as follows:

- We propose a novel causal framework and formalize the FICIM and PICIM causal models for domain adaptation. This causal framework and model can provide a solid theoretical foundation for domain adaptation problems.
- We propose a causal bound for the OSDA. This bound can guide the development of new algorithms for OSDA problems.
- We prove that when the source domain follows the FICIM, ERM is sufficient for model training. Our work demonstrates the feasibility of constructing artificial FICIM datasets instead of natural datasets for training.
- Our theoretical work on domain adaptation can guide the generation of diverse and representative training datasets using LLMs, enhancing model generalization and adaptability through a focus on causal relationships and data selection. Additionally, our theory can guide the selection of high-quality datasets for efficient pre-training and fine-tuning of LLMs.

## 2 Related work

In this section, we first introduce OSDA and Open Set Recognition (OSR) theories. Then, we review DA from a causal view. For detailed information on related works, please refer to Appendix B.

### 2.1 OSDA and OSR theories

Our research problem is within the field of OSDA. A similar concept related to OSDA is OSR (Geng et al., 2021). Hence we refer readers to (Geng et al., 2021; Yang et al., 2024) for comprehensive surveys of OSDA and OSR. Early theoretical studies on OSR formalized the relationship between the known and unknown classes using the open space risk (Wang et al., 2023; Rastegar et al., 2024) and extreme value theory (Petit et al., 2023), but they did not provide theoretical guarantees. Moreover, none of the above-mentioned works can solve our problem because they need a strict assumption that at least one observed distribution does not change across domains.

---

[1]As shown in Fig. 1, the information that determines the image label is solely the shape of the digit in the image, not the background color. Changing the color does not affect the image label.

## 2.2 DA FROM A CAUSAL VIEW

Existing studies primarily assume invariant predictors or rely on different causal assumptions to address domain adaptation problems (Magliacane et al., 2018; Li et al., 2024). Although Invariant Risk Minimization (IRM) (Liu et al., 2024) methods are commonly used for learning robust representations, research has shown that they do not necessarily outperform ERM (Rosenfeld et al., 2021; Buchholz et al., 2024). Despite some success with these methods (Chen & Bühlmann, 2021; Sun et al., 2021; Liu et al., 2021; Huang et al., 2024), they fail to provide a theoretical understanding of nonlinear high-dimensional data, and only consider a variation of the PICIM in our work as their causal structure, whereas we consider the FICIM and PICIM.

## 3 A CAUSAL FRAMEWORK OF DOMAIN ADAPTATION

### 3.1 NOTATIONAL PRELIMINARIES

We denote $\Omega$, $\mathscr{A}$, and $\mathbb{P}$ as the original sample space, $\sigma-$algebra on $\Omega$, and probability measure, respectively. Then, $(\Omega, \mathscr{A}, \mathbb{P})$ is a probability space. We use capital letters such as $X$ to denote random elements and boldface letters such as $\mathbf{x}$ to denote value vectors. Calligraphic capital letters such as $\mathcal{X}$ are used for space. Random elements are measurable maps. for instance, $X : (\Omega, \mathscr{A}) \rightarrow (\mathcal{X}, \mathscr{B})$. For simplicity, we use notations including $\mathbb{P}_X$, $\mathbb{P}_{XY}$, and $\mathbb{P}_{X|Y}$ to denote the marginal, joint, and conditional distributions, respectively. Moreover, $p$ is the probability density function. For more symbol annotations and terminology, see Table 5 in Appendix A.

### 3.2 CAUSAL ASSUMPTIONS

Causality research indicates that real-world data distributions stem from underlying causal mechanisms that are typically invariant across domains (Pearl & Mackenzie, 2018; Schölkopf, 2022). Liu et al. (2021) formalized this into the *causal invariance principle*, asserting that causal generation mechanisms remain consistent across different domains. For high-dimensional data $X$—such as text, images, or audio—it's commonly assumed that $X$ is a nonlinear function of latent attributes $A$ (Locatello et al., 2019; 2020; Von Kügelgen et al., 2021). However, not all attributes in $A$ are invariant causes of $X$ or the target label $Y$. Some attributes, like noise or background color, may affect $X$ but not $Y$. Therefore, we partition $A$ into two subsets: the causally invariant attributes $C$ and the variation attributes $V$, where $C$ maintains invariant relationships with both $X$ and $Y$. We formalize this with the following assumption:

**Assumption 1.** *(Causal invariance assumption) For high-dimensional data $X$ and its prediction target $Y$, the latent attribute set $A$ between $X$ and $Y$ can be divided into the causally invariant attribute set $C$ and variation attribute set $V$. Attributes belonging to $C$ should satisfy $\mathbb{P}(Y|C)$ and $\mathbb{P}(X|C)$ being invariant across domains. Attributes belonging to $V$ should satisfy that $\mathbb{P}(Y|V)$ or $\mathbb{P}(X|V)$ varies across domains.*

Based on this, we define:

**Definition 1.** *(Probability Generation Model (PGM)). We define the probability generation model for high-dimensional data as certain statistical probability descriptions of the data generation process, i.e., $PGM = \langle \mathbb{P}_C, \mathbb{P}_{X|C}, \mathbb{P}_{Y|C}, \mathbb{P}_{V|C}, \mathbb{P}_{YVX} \rangle$ on high-dimensional data $X$ and target $Y$ with causally invariant attributes $C$ and variation attributes $V$.*

This assumption is supported across various fields (Liang et al., 2018; Yue et al., 2021). For example, in computer vision, images from the same class share causally invariant attributes, while variation attributes provide class-independent features like color and background (Liang et al., 2018).

By constructing a probability model with separated latent attributes, we can better describe the data generation process. For example, when high-dimensional data is collected from different sensors with varying characteristics, the PGM accounts for variations by incorporating causally invariant attributes (e.g., physical properties) and variation attributes (e.g., sensor-specific features).

### 3.3 INVARIANT CONNECTIONS ACROSS DOMAINS

Traditional domain definitions focus on observed distributions or labeling functions associated with $X$ and $Y$ (Ben-David et al., 2010; Fang et al., 2020), which may not capture essential differences involving latent attributes in high-dimensional data. Instead, we aim to mathematically characterize the invariant relationship between the source and target domains. Therefore, to address this issue, we define:

**Definition 2.** *(Domain). A domain $d$ includes a series of observed data distributions $\mathbb{P}^d$ that are generated by a domain-specific $PGM_d = \langle \mathbb{P}^d_C, \mathbb{P}^d_{X|C}, \mathbb{P}^d_{Y|C}, \mathbb{P}^d_{V|C}, \mathbb{P}^d_{YVX} \rangle$. For the convenience of subsequent use, we construct a domain set $D = \{d_1, d_2, ...\}$ where every domain $d_i \in D$ is generated by $PGM_{d_i}$.*

This definition allows us to express invariant relationships between domains:

**Proposition 1.** *(Domain invariance) Given two arbitrary domains $d_i, d_j \in D$, we have an invariant relationship $\mathbb{P}^{d_i}_{X|C} = \mathbb{P}^{d_j}_{X|C}$ and $\mathbb{P}^{d_i}_{Y|C} = \mathbb{P}^{d_j}_{Y|C}$.*

*Proof.* Using Definition 2, we can directly derive the domain invariance proposition from Assumption 1. $\square$

To address the unknown relationship between $Y$ and $V$, we introduce two causal models shown in Fig. 2:

**Definition 3.** *(Fully Informative Causal Invariance Model (FICIM)) A causal model is considered to be FICIM if $\mathbb{P}_{Y|C}$ is invariant across domains and the relationship between $Y$ and $V$ is not present or not relevant.*

**Definition 4.** *(Partially Informative Causal Invariance Model (PICIM)) A causal model is considered to be PICIM if $\mathbb{P}_{Y|C}$ varias across domains and there exists an unknown or uncertain relationship between $Y$ and $V$.*

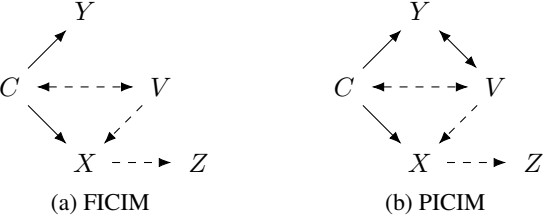

(a) FICIM      (b) PICIM

Figure 2: The causal graph structure of FICIM and PICIM.

**Discussion.** In FICIM, $C$ influences both $X$ and $Y$, while $V$ may affect $X$ but not $Y$. In PICIM, $C$ still influences $X$ and $Y$, but $V$ may also have an effect on $Y$. Unlike the Fully Informative Invariant Features (FIIF) and Partially Informative Invariant Features (PIIF) in (Ahuja et al., 2021), our model focuses on the latent attributes $C$ and $V$ that are fundamental to data generation, distinguishing $C$ from $V$ based on domain invariance. In FICIM, $\mathbb{P}_{Y|C,V} = \mathbb{P}_{Y|C}$; in PICIM, $\mathbb{P}_{Y|C,V} \neq \mathbb{P}_{Y|C}$:

**Proposition 2.** *(Properties for causal diagrams) If a domain $d_i \in D$ follows the FICIM, $\mathbb{P}^{d_i}_{Y|CV} = \mathbb{P}^{d_i}_{Y|C}$. If a domain $d_i$ follows the PICIM, $\mathbb{P}^{d_i}_{Y|CV} \neq \mathbb{P}^{d_i}_{Y|C}$.*

*Proof.* Obviously, the proposition can be obtained from the causal Markov condition (Janzing & Schölkopf, 2010). $\square$

We formalize the invariant connections as:

**Theorem 1.** *(Invariant connections for causal information of observed data) Given two arbitrary domains $d_i, d_j \in D$ following the FICIM or PICIM, there exists a random element $X^C$ that can*

be sampled from a function $f : \mathcal{C} \times [0, 1] \to \mathcal{X}$ and $T$ follows a uniform distribution $U[0, 1]$; i.e., $X^C = f(C, T)$, such that :

$$p_{Y|X^C}^{d_i}(\mathbf{y}|\mathbf{x}) = p_{Y|X^C}^{d_j}(\mathbf{y}|\mathbf{x}).$$

This theorem indicates that focusing on the causal information derived from invariant attributes $C$ allows us to establish invariant predictive relationships across domains.

## 4 PROPOSED BOUND FOR OSDA

### 4.1 MOTIVATION AND DEFINITIONS

In OSDA, the primary goal is to train a classifier using data from a source domain that can accurately identify known classes and distinguish between known and unknown classes in a target domain. We focus on the challenging case of a single-source domain problem. Specifically, we consider a source domain $S$ and a target domain $T$ from the domain set $D$, i.e., $S, T \in D$, satisfying the properties defined in Definition 2.

**Motivation problem.** *(Learning for OSDA). Given a single source domain $S \in D$ and a target domain $T \in D$, we observe a training dataset $\mathbf{D}_S = \{(\mathbf{x}_i, \mathbf{y}_i)\}_{i=1}^{N_S}$ that is obtained from $S$ and the label space $\mathcal{Y}_S \subset \mathcal{Y}_T$ is known; that is, the testing samples in the target domain belong to unknown classes that do not appear in the source domain. The goal is to identify a known class label and to separate known samples from unknown samples in the target domain $T$.*

To address this problem, we need a theoretical risk decomposition for the target domain. We define the expected risk as follows:

**Definition 5.** *(Expected risk conditional on the domain). Given a random element $Y$ and a fitted element $\hat{Y}$ of space $\mathcal{Y}$ from domain $d_i \in D$, we formulate the following definitions for an arbitrary loss function $\ell : \mathcal{Y} \times \mathcal{Y} \to \mathbb{R}^+$ :*

*1. The expected risk is defined as:*

$$R_{d_i}^{\ell}(Y) := \mathbb{E}_{P_Y^{d_i}} \ell(\hat{\mathbf{y}}, \mathbf{y}).$$

*For simplicity, we omit $\ell$ in the subsequent form of $R_{di}^{\ell}$, that is, $R_{di} := R_{di}^{\ell}$.*

*2. Given a random element $U$ of space $\mathcal{U}$ that is jointly distributed with $Y$ with an arbitrary mapping $\psi : \mathcal{U} \to \mathcal{Y}$, the quantity*

$$R_{d_i}(Y|U) := \mathbb{E}_{P_{YU}^{d_i}} \ell(\psi(\mathbf{u}), \mathbf{y}).$$

*3. Given $R_{d_i}(Y|U)$, the minimum expected risk of predicting $Y$ given $U$ is*

$$R_{d_i}^*(Y|U) := \inf_{\psi} R_{d_i}(Y|U).$$

**Definition 6.** *(Stable expected risk conditional on domain). Given a domain $d_i \in D$, the stable expected risk is defined as:*

$$R_{d_i}(Y|X^C) = \mathbb{E}_{\mathbf{P}_{X^C Y}^{d_i}} \ell(f(\mathbf{x}), \mathbf{y}),$$

*where $X^C$ is the causal information of the observed data that can be sampled from a function $f : \mathcal{C} \times [0, 1] \to \mathcal{X}$ satisfying*

$$p_{X^C}(\mathbf{x}) = p_C(\mathbf{c}) \qquad \forall \mathbf{c} \in \mathcal{C}.$$

**Discussion.** According to Proposition 1, the relationship between the causal attributes $C$ and $Y$ is stable across domains. While the complete set of latent attributes $CV$ provides sufficient information for predicting $Y$, relying on them may not yield stable minimum risk in the target domain due to the instability of $P_{Y|CV}$ without label information from the target domain. To achieve a stable minimum risk, we focus on the invariant causal connections derived from the observed data, specifically using the causal component $X^C$ associated with $C$.

## 4.2 THEORETICAL RESULTS

This section presents the main theoretical results, which not only emphasize the importance of causal invariance in achieving effective domain adaptation but also provide bounds on the stable expected risk in both closed-set and open-set domain adaptation scenarios. Full proofs are included in Appendix C.

**Theorem 2.** *(Theoretical bound of stable expected risk under closed-set domain adaptation). Given a single source domain $S \in D$ and a target domain $T \in D$, and further assuming the label space $\mathcal{Y}_S = \mathcal{Y}_T = \mathcal{Y}$, we obtain a theoretical bound of the semantic controlled risk $R_T(Y|X^C)$, where $X^C$ can be sampled from a function $f : \mathcal{C} \times [0,1] \to \mathcal{X}_T$ as follows:*

$$R_T(Y|X^C) \leq (1+\beta)R_S(Y|X^C),$$

*where $\beta = \sup C \in \mathcal{C}p^T C(\mathbf{c})/p^S C(\mathbf{c}) - 1$, under the condition that there exist positive constants $0 < m \leq M$ such that $m \leq p^T C(\mathbf{c}), p_C^S(\mathbf{c}) \leq M$ for all $\mathbf{c} \in \mathcal{C}_S$.*

> **Intuition.** This result builds upon existing works focusing on generalization bounds in closed-set domain adaptation, such as ERM, causal conditional shift, and discrepancy distance. Our approach innovatively emphasizes the importance of causal relationships and invariance in the data generation process, contributing to enhancing domain adaptation capabilities in practical applications.

**Theorem 3.** *(Theoretical bound of stable expected risk under OSDA). Given a single source domain $S \in D$ and a target domain $T \in D$, and further assuming the label space $\mathcal{Y}_S \subset \mathcal{Y}_T$ and setting $\mathbf{y}^{uk}$ to represent the unknown target classes $\mathcal{Y}_T \setminus \mathcal{Y}_S$, we can obtain a theoretical bound of the stable expected risk $R_T(Y|X^C)$ under OSDA, as follows:*

$$R_T(Y|X^C) \leq \underbrace{(1+\beta)R_S(Y|X^C)}_{\text{(1) Risk of known target classes}} + \underbrace{\int_{\mathcal{X}} \ell(f(\mathbf{x}), \mathbf{y}^{uk})p_{X^C Y}^T(\mathbf{x}, \mathbf{y}^{uk})d\mathbf{x}}_{\text{(2) Risk of unknown target classes}},$$

*where $\beta = \sup_{C \in \mathcal{C}_S} p_C^T(\mathbf{c})/p_C^S(\mathbf{c}) - 1$, under the condition that there exist positive constants $0 < m \leq M$ such that $m \leq p^T C(\mathbf{c}), p_C^S(\mathbf{c}) \leq M$ for all $\mathbf{c} \in \mathcal{C}_S$.*

> **Intuition.** The bound in Theorem 3 consists of two terms: the risk of known target classes and the risk of unknown target classes. This decomposition clarifies the components of the stable expected risk in the target domain and guides the minimization process.

**Remark 1.** *For Theorem 3, the second term of the bound $\int_{\mathcal{X}} \ell(f(\mathbf{x}), \mathbf{y}^{uk})p_{X^C Y}^T(\mathbf{x}, \mathbf{y}^{uk})d\mathbf{x}$ can be minimized by the optimal representation $Z_S^* = \varphi_S^*(X)$ that is obtained by the ERM of a source domain $S$:*

$$\inf_{f \in \mathcal{F}} \int_{\mathcal{X}} \ell(f(\mathbf{x}), \mathbf{y}^{uk})p_{X^C Y}^T(\mathbf{x}, \mathbf{y}^{uk})d\mathbf{x}$$

$$= \inf_{g \in \mathcal{G}} \int_{\mathcal{X}} \ell(g(\varphi_S^*(\mathbf{x})), \mathbf{y}^{uk})p_{X^C Y}^T(\mathbf{x}, \mathbf{y}^{uk})d\mathbf{x}$$

*with the assumption that*

$$\mathcal{F} = \mathcal{G} \circ \Phi \qquad \forall f \in \mathcal{F}, g \in \mathcal{G}, \varphi \in \Phi.$$

> **Intuition.** The bound in Theorem 3 suggests that a good model for handling OSDA should i) seek a classifier $f_S^* = g_S^* \circ \varphi_S^*$ that minimizes the stable expected risk $R_S(Y|X^C)$ of the source domain, and ii) determine an optimal open set classifier $g_T^*$ for separating the knowns and unknowns based on the representations $\varphi_S^*(X_T)$.

Next, we will discuss under what conditions performing ERM solely in the source domain is sufficient.

**Theorem 4.** *For Theorem 3, conducting the ERM on a FICIM source domain can provide enough information to bound the stable expected risk of the target domain $R_T(Y|X^C)$.*

> **Intuition.** Since the causal attributes $C$ contain all information about the outcome variable $Y$, and the variation $V$ adds no extra information about $Y$, the ERM on the source domain yields $R_S^*(Y|X) = R_S^*(Y|X^C)$. Consequently, ERM on a FICIM source domain provides sufficient information to find an optimal open-set classifier, adequately bounding the stable expected risk $R_T(Y|X^C)$ in the target domain.

**Theorem 5.** *For Theorem 3, conducting the ERM on a PICIM source domain cannot bound the stable expected risk of target domain $R_T(Y|X^C)$.*

> ***Intuition.*** *The key distinction between the theorem of the PICIM source domain and Theorem 4 lies in the capacity of the variation $V$ to predict $Y$ within causal models, as the additional information in $V$ may be detrimental in the target domain. Consequently, the expected risk $R_T(Y|X_C)$ in the target domain is contingent upon the amount of additional information in $V$, with a preference for scenarios where $V$ contains less additional information.*

**Remark 2.** *Given a target domain $T \in D$, we obtain a useful decomposition of the minimum expected risk as follows:*

$$R_T^*(Y|X) = \underbrace{R_T^*(Y|X^C)}_{\text{(1) Minimum stable expected risk}} - \underbrace{[R_T^*(Y|X^C) - R_T^*(Y|X^{CV})]}_{\text{(2) Uncontrollable spurious benefit}}.$$

> **Intuition.** This remark is straightforward: minimizing $R_T^*(Y|X)$ is equivalent to minimizing $R_T^*(Y|X^{CV})$. By adding and subtracting $R_T^*(Y|X^C)$, we see that the target risk equals the minimum stable expected risk minus the spurious benefit from variation information. Since this benefit is independent of the source domain, it's reasonable to replace the objective of minimizing the total expected risk with that of minimizing the stable expected risk.

## 5 EXPERIMENTS

First, we conducted comprehensive OSDA and OOD tasks[2] on the CM-NIST dataset to validate our proposed theory. Next, we performed experiments on synthetic data, showcasing

Table 1: Description of all our tasks.

| Input (X) | Label (Y) | Variation attribute (V) | Invariant attribute (C) |
|---|---|---|---|
| Cmnist | $\{0,1\}$ | Color | Digit |
| Synthetic data | $\{0,1\}$ | $\{0,\ldots,7\}$ | - |
| Restaurant review | Restaurant rating | Food-mention | Service, Noise, Ambiance, Food |

different special cases, all of which are explained by our unified theoretical framework, demonstrating the applicability of our theoretical results. Finally, we conducted experiments on restaurant review (text) data and applied our theoretical findings to instruction fine-tuning of large models. These experiments fully demonstrate our two final theoretical results: 1) performing ERM on a FICIM source domain provides enough information to bound the stable expected risk of the target domain ((Theorem 4)), and 2) performing ERM on a PICIM source domain cannot bound the stable expected risk of the target domain (Theorem 5). Table 1 provides an overview of the tasks we experiment with.

### 5.1 OS-CMNIST DATASET

**Experimental Setup.** We constructed our open-set CMNIST (OS-CMNIST) dataset following the dealing method of CMNIST (Arjovsky et al., 2020) to satisfy the setting demand of the OSDA and OOD detection task. To ensure a fair comparison, we adopted the same loss function and model architecture as used in existing studies (Chen et al., 2021). To fully demonstrate the validity of

---

[2]OSDA primarily focuses on how to handle these unknown categories in the target domain while maintaining good performance on known categories, whereas OOD detection emphasizes distinguishing between known and unknown categories without necessarily involving the specific learning of classes.

our theoretical results, we constructed two sets of data ( FICIM group and PICIM group). For the FICIM group, the key parameters were $Corr(V,C)_S = 0.8$, $Corr(Y,C)_S = 1$, $Corr(V,C)_T = 0.1$. For the PICIM group, the key parameters were $Corr(V,C)_S = 0.8$, $Corr(Y,C)_S = 0.75$, $Corr(V,C)_T = 0.1$. For detailed experimental setup, results, and analysis, please refer to Appendix D.1.

**Results.** As indicated in Table 2, training with ERM on the FICIM source domain under different loss functions could achieve nearly perfect performance for the CS-ACC, AUROC, and OSCR of the target domain, which supports Theorem 4. We demonstrate Theorem 5 by observing that the CS-ACC, AUROC, and OSCR declined sharply from the FICIM source domain to the PICIM source domain. That is, training with ERM on the PICIM source domain resulted in a model that performed worse on the target domain than on the FICIM source domain.

Table 2: Performance comparison of FICIM and PICIM source domain on CMNIST.

| Method | | CS-ACC | AUROC | OSCR |
|---|---|---|---|---|
| ARPLoss (Chen et al., 2021) | FICIM | $99.63 \pm 0.01$ | $95.52 \pm 1.01$ | $95.38 \pm 1.03$ |
| | PICIM | $64.30 \pm 0.32$ | $52.22 \pm 1.19$ | $42.37 \pm 0.46$ |
| ARPLoss+CS (Chen et al., 2021) | FICIM | $99.66 \pm 0.04$ | $96.47 \pm 0.22$ | $96.34 \pm 0.25$ |
| | PICIM | $67.69 \pm 0.82$ | $53.07 \pm 0.70$ | $40.93 \pm 0.58$ |
| RPLOSS (Chen et al., 2020) | FICIM | $99.51 \pm 0.01$ | $91.65 \pm 2.95$ | $91.48 \pm 2.94$ |
| | PICIM | $64.23 \pm 2.00$ | $53.76 \pm 1.75$ | $45.05 \pm 0.70$ |
| Softmax | FICIM | $99.48 \pm 0.01$ | $94.20 \pm 0.55$ | $94.03 \pm 0.54$ |
| | PICIM | $62.15 \pm 0.99$ | $52.81 \pm 1.59$ | $43.23 \pm 0.57$ |
| GCPL (Yang et al., 2020) | FICIM | $99.60 \pm 0.01$ | $95.98 \pm 0.17$ | $95.84 \pm 0.18$ |
| | PICIM | $66.10 \pm 2.71$ | $53.02 \pm 2.71$ | $43.67 \pm 1.35$ |

## 5.2 SYNTHETIC DATA

**Experimental Setup.** To further validate the effectiveness of our theoretical framework, we conducted experiments on synthetic data. Following the experimental setups of existing studies (Feder et al., 2023), we generate synthetic data for a binary classification problem where $|V| = 8$ (cardinality of varying attribute V). We sample $\mathbb{P}(V|Y)$ to simulate varying degrees of spurious correlations. Then we draw $x = [x^*, x_{\text{spu}}]$ from a Gaussian distribution,

$$x_i = \begin{bmatrix} x^* \\ x_{\text{spu},i} \end{bmatrix} \backsim \mathcal{N}\left( \begin{bmatrix} \mu_{y_i} \\ \mu_{c_i} \end{bmatrix}, \begin{bmatrix} \sigma^2 \mathbf{I}_{\mathbf{d}^*} & 0 \\ 0 & \sigma_{\text{spu}}^2 \mathbf{I}_{\mathbf{d_c}} \end{bmatrix} \right) ,$$

In this case the counterfactual $\hat{x}_i(v)$ for the sample $x_i$ is obtained by adding $\mu_v - \mu_{v_i}$ to $x_{\text{spu},i}$. To corrupt our augmentation, we instead add $\xi_i(\mu_v - \mu_{v_i})$ where $\xi_i$ is drawn from a truncated Gaussian centered at $\lambda \in (0,1)$. We train models with a fixed sample size and evaluate the trained models' performance on unconfounded distribution $P_\perp$ to examine the interplay between spurious correlation strength (measured by mutual information $I(Y;V)$). Different mutual information values $I(Y;V)$ represent varying degrees of PICIM source domain. When the mutual information is zero, it indicates FICIM source domain. For detailed experimental setup, results, and analysis, please refer to Appendix D.2.

**Results.** As shown in Fig. 3, under different corruptions, the model's performance decreases. Compared to corruptions, spurious correlations have a greater impact on the model's performance. This further demonstrates that training with ERM on the PICIM source domain results in worse performance on the target domain compared to the FICIM source domain. Moreover, by employing certain augmentation techniques and methods, modifying the training mechanism of the model can mitigate the differences caused by the two data generation processes.

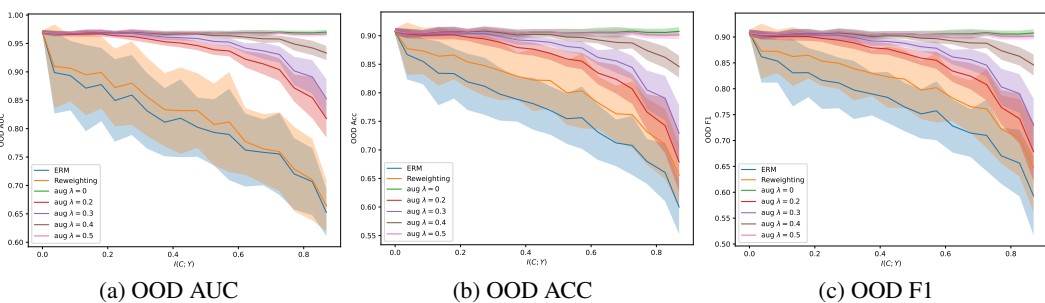

|                | (a) OOD AUC | (b) OOD ACC | (c) OOD F1 |
|----------------|-------------|-------------|------------|

Figure 3: Model performance with different parameter settings on synthetic data. Lower values of $\lambda$ correspond to stronger corruptions of the augmentations.

### 5.3 RESTAURANT REVIEWS DATA

**Experimental Setup.** We use the CEBaB dataset (Abraham et al., 2022), which consists of short restaurant reviews and ratings from OpenTable[3], including evaluations for food, service, noise, ambiance, and an overall rating. We used the train-exclusive split of the dataset, which contains 1, 755 examples. We focus on an experimental setup: a modified version called CeBAB-Spurious, where there is a spurious correlation between the labels $Y$ and variable attributes $V$.

To construct CeBAB-Spurious, we leveraged the availability of both the original and perceived ratings for each review in CeBAB. The original rating represents the reviewer's initial thoughts when writing the review, while the perceived rating indicates whether the review contains information about various restaurant attributes (e.g., food, service, noise, ambiance) and their associated sentiment. We utilized this unique data structure to capture reviewers' writing styles. Some reviewers are concise and provide limited descriptions, while others are more detailed and include more information. Inspired by existing research (Feder et al., 2023), we introduced a new attribute called food-mention to signify the presence of food-related information in a review. If the perceived food rating is either negative or positive, we assign a value of 1 to the food-mention attribute; otherwise, it is set to 0. We sample the data such that the correlation between food-mention and outcomes is 0.45. Please note that the sampled data follows the PICIM, while the data from counterfactual interventions using GPT-4 (Achiam et al., 2023) follows the FICIM. For detailed experimental setup, results, and analysis, please refer to Appendix D.3.

**Results.** As shown in Table 3 and Table 4, when debiasing different restaurant features, our theoretical results effectively explain the model's performance differences under various data generation mechanisms. The main conclusions include the following two points: (1) Based on common knowledge, restaurant noise has a causal relationship with overall restaurant ratings. In this case, when we debias for restaurant noise, the model is unable to leverage these useful causal signals, leading to an increase in the minimum stable expected risk. Essentially, by removing the noise, the model can no longer capture the useful information embedded in it, resulting in reduced stability in its predictions. (2) Food mention is a spurious feature, meaning it has no direct causal relationship with restaurant ratings. By debiasing for food mentions, the model eliminates the influence of irrelevant, spurious correlations. This helps improve the model's performance, as it can focus more on the true causal signals relevant to the task, without being distracted by unrelated features.

### 5.4 EFFICIENT FINE-TUNING

**Experimental Setup.** To further validate that our theory can guide the selection of high-quality data for efficient pre-training and fine-tuning of large models, we construct instruction pairs based on restaurant reviews to fine-tune different large models (LLaMA3-8B, ChatGLM4-9B, and Qwen2-7B [4]). For specific experimental setting, the construction of instruction pairs, and more results, please refer to the Appendix D.3.

---

[3]https://www.opentable.com/
[4]https://modelscope.cn/home.

Table 3: Performance comparison of FICIM and PICIM source domain on food-mention of restaurant reviews.

| Methods | | ACC | F1 | Precision |
|---|---|---|---|---|
| $Corr(V,Y) = 0.45$ | FICIM | $72.00 \pm 0.01$ | $71.53 \pm 0.20$ | $71.60 \pm 0.47$ |
| | PICIM | $67.00 \pm 3.01$ | $63.79 \pm 3.38$ | $67.21 \pm 2.77$ |
| $Corr(V,Y) = 0.40$ | FICIM | $70.04 \pm 0.75$ | $67.46 \pm 2.17$ | $67.08 \pm 2.26$ |
| | PICIM | $68.09 \pm 1.02$ | $64.41 \pm 2.38$ | $65.94 \pm 1.91$ |
| $Corr(V,Y) = 0.35$ | FICIM | $70.32 \pm 1.04$ | $67.91 \pm 0.79$ | $71.53 \pm 0.05$ |
| | PICIM | $68.13 \pm 2.12$ | $65.56 \pm 0.95$ | $68.41 \pm 2.69$ |
| $Corr(V,Y) = 0.30$ | FICIM | $70.00 \pm 2.36$ | $67.07 \pm 2.19$ | $73.21 \pm 2.62$ |
| | PICIM | $67.00 \pm 1.24$ | $64.95 \pm 1.75$ | $67.50 \pm 0.91$ |

Table 4: Performance comparison of FICIM and PICIM source domain on Restaurant noise of restaurant reviews.

| Methods | | ACC | F1 | Precision |
|---|---|---|---|---|
| $Corr(C,Y) = 0.45$ | FICIM | $79.00 \pm 2.61$ | $78.41 \pm 3.92$ | $78.27 \pm 2.46$ |
| | PICIM | $66.00 \pm 1.60$ | $67.97 \pm 2.69$ | $71.08 \pm 2.77$ |
| $Corr(C,Y) = 0.40$ | FICIM | $70.00 \pm 1.51$ | $69.78 \pm 2.47$ | $69.53 \pm 0.85$ |
| | PICIM | $64.13 \pm 0.60$ | $65.00 \pm 2.41$ | $67.24 \pm 4.06$ |
| $Corr(C,Y) = 0.35$ | FICIM | $79.50 \pm 1.35$ | $79.31 \pm 0.43$ | $79.27 \pm 3.07$ |
| | PICIM | $71.00 \pm 1.55$ | $66.64 \pm 2.28$ | $65.08 \pm 2.97$ |
| $Corr(C,Y) = 0.30$ | FICIM | $75.32 \pm 4.06$ | $74.89 \pm 3.19$ | $74.27 \pm 3.42$ |
| | PICIM | $72.11 \pm 3.41$ | $69.46 \pm 5.05$ | $69.64 \pm 1.29$ |

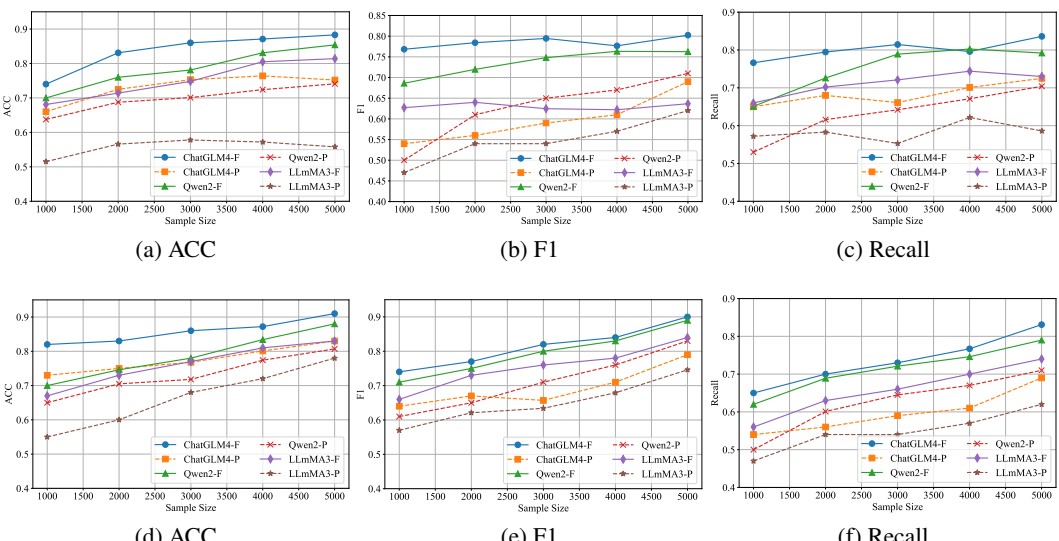

(a) ACC  (b) F1  (c) Recall

(d) ACC  (e) F1  (f) Recall

Figure 4: Performance comparison of fine-tuning based on food-mention (the upper part of the figure) and restaurant reviews (the lower part of the figure). $-$P indicates fine-tuning based on the PICIM source domain, while $-$F indicates fine-tuning based on the FICIM source domain.

**Results.** As shown in Fig. 4, as the amount of fine-tuning data increases, the model performance improves. The growth trends in performance vary for different large language models. Relatively speaking, the performance of LLaMA3 is somewhat inferior in our task. More importantly, under the FIFCM source domain, fine-tuning GLM with 2,000 samples achieves performance comparable to fine-tuning Qwen2 and LLaMA3 with 5,000 samples. Additionally, in GLM, under the FIFCM mechanism, 1,000 samples can achieve the performance obtained after fine-tuning the model with 5,000 samples under the PICIM mechanism. This further indicates that our theoretical results can guide the efficient fine-tuning or even pre-training of LLMs.

# 6 CONCLUSIONS

We have proposed a causal bound for OSDA of high-dimensional data. Using this bound, we theoretically proved that the FICIM and PICIM source domains can explain the performance difference of ERM: (1) The ERM when the source domain follows FICIM can provide sufficient information to bound the stable expected risk of the target domain. (2) The ERM when the source domain follows PICIM cannot bound the stable expected risk of the target domain. We demonstrated the effectiveness of our theoretical results by conducting comparative experiments on FICIM and PICIM datasets, and showed that state-of-art open-set algorithms performed poorly when only the PICIM dataset was used. Our theoretical and experimental results revealed the limitation of existing algorithms for OSDA, including OSR. We anticipate that our study may pave the way for new algorithm designs for OSDA and other simpler domain adaptation challenges, as it provides fundamental knowledge for these problems.

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

APPENDIX

# Contents

## A  NOTATION AND TERMINOLOGY

| Symbol | Description |
|--------|-------------|
| $\mathcal{N}$ | The Gaussian distribution |
| $\mathbf{I}$ | Identity matrix |
| $\mathrm{x}^*$ | Core feature |
| $\mathrm{x}_{\mathrm{spu}}$ | Spurious feature |
| $Corr$ | The correlation coefficient between two variables |
| $\kappa_S$ | The correlation coefficient between two variables on source domain |
| $\mathcal{F}$ | Function space |
| $N_S$ | The number of samples in the training set |
| $\lambda$ | The noise intensity |

Table 5: Symbol Notations and Their Descriptions

## B  RELATED WORK

In this section, we first introduce OSDA and OSR theories. Then, we review domain adaptation from a causal view.

### B.1  OSDA AND OSR THEORIES

Our research problem is within the field of OSDA. A similar concept related to OSDA is OSR (Geng et al., 2021). Hence we refer readers to (Geng et al., 2021) for comprehensive surveys of OSDA and OSR. Early theoretical studies on OSR formalized the relationship between the known and unknown classes using the open space risk (Scheirer et al., 2014; Wang et al., 2023; Rastegar et al., 2024) and extreme value theory (Rudd et al., 2017; Petit et al., 2023), but they did not provide theoretical guarantees.

Liu et al. (2018) provided the sample complexity for guaranteeing the detection rate of OSR, whereas Fang et al. (2021) proposed a generalization bound for OSR based on the PAC theory, which demonstrated the theoretical existence of an OSR algorithm. Zhang et al. (2020) constructed an unbiased risk estimator by exploiting unlabeled training data to approximate the underlying distribution of the unknown classes. These bounds provided by (Liu et al., 2018; Fang et al., 2021; Zhang et al., 2020) highlight the need to find an augmented domain to represent a group of novel classes.

The class-incremental domain adaptation paradigm proposed by (Kundu et al., 2020), which is almost the same as the OSDA, derived a bound for the target-domain risk by considering the target-shared risks and target-private risks independently. Fang et al. (2020) proposed a theoretical bound for the OSDA problem first. However, these bounds in (Kundu et al., 2020) and (Fang et al., 2020) can not explain why or when ERM performs well for OSDA.

Moreover, none of the above-mentioned works can solve our problem because they need a strict assumption that at least one observed distribution does not change across domains.

## B.2 DA FROM A CAUSAL VIEW

The framework of Structural Causal Models (SCMs) (Pearl, 2009) has motivated many interesting works on causal discovery and causal inference. Inspired by SCMs, our work mainly uses the principle of Invariant Causal Prediction (ICP). ICP considers the invariance of the conditional distribution of the target variable Y given its direct causes, which has been articulated numerous times (Pearl, 2009; Pearl & Mackenzie, 2018; Peters et al., 2016; Rojas-Carulla et al., 2018; Pfister et al., 2019) and has been formulated by (Peters et al., 2016; Heinze-Deml et al., 2018). Rojas-Carulla et al. (2018); Magliacane et al. (2018); Li et al. (2024) relate domain adaptation with the invariant causal prediction principle, which inspires our work. These works mentioned above mostly assume that the observed predictors or a subset of observed predictors are causally invariant.

Although IRM (Arjovsky et al., 2020; Liu et al., 2024) methods aim to learn robust and invariant representations, recent studies have shown that they may not always outperform the ERM objective (Nagarajan et al., 2020; Rosenfeld et al., 2021; Buchholz et al., 2024). This observation highlights the need for a deeper understanding of the performance and limitations of both IRM and ERM in the context of high-dimensional data and domain adaptation problems.

The concept of our causal framework is similar to that of (Cai et al., 2019; Chen & Bühlmann, 2021; Sun et al., 2021; Liu et al., 2021; Huang et al., 2024; Carvalho et al., 2024). In comparison, Cai et al. (2019) assumed the independence of the latent semantic variables and other latent variables, which differs from our dependence assumption. Chen & Bühlmann (2021) adopted linear structural causal models to study complicated domain adaptation problems, which did not provide a theoretical understanding of nonlinear high-dimensional data. Sun et al. (2021); Liu et al. (2021) only considered a variation of the PICIM in our work as their causal structure, whereas we consider the FICIM and PICIM.

Moreover, the above methods do not solve the OSDA problem, which is the crucial problem of our study.

## C THEORETICAL PROOF

We begin with an important lemma by following Lemma 3.22 of (Kallenberg, 2002) to prove Theorem A.1.

**Lemma A.1.** *(Kallenberg, 2002) Given random variables $X \in \mathcal{X}$, $C \in \mathcal{C}$ from a domain $d_i \in D$; i.e., the Markov chain is $X - C$, there exists a random element $X^C$ that can be sampled from a function $f : \mathcal{C} \times [0, 1] \to \mathcal{X}$. Furthermore, $T$ follows a uniform distribution $U[0, 1]$; i.e., $X^C = f(C, T)$, such that the joint probability density function of $X^C$ and $C$ satisfies*

$$p_{X^C,C}^{d_i}(\mathbf{x}, \mathbf{c}) = p_{X,C}^{d_i}(\mathbf{x}, \mathbf{c}) \quad \text{for any } (\mathbf{x}, \mathbf{c}) \in \mathcal{X} \times \mathcal{C}.$$

*([Kallenberg](), [2002]()) presents the proof for the above conclusion. As $X \in \mathcal{X}$ and $X^C \in \mathcal{X}$, we can obtain another property for $X^C$,*

$$p_{X^C|C}^{d_i}(\mathbf{x}|\mathbf{c}) = p_{X|C}^{d_i}(\mathbf{x}|\mathbf{c}) \quad \text{for any } (\mathbf{x}, \mathbf{c}) \in \mathcal{X} \times \mathcal{C}.$$

**Theorem A.1.** *([Theorem 1]()) Given two arbitrary domains $d_i, d_j \in D$ following the FICIM or PICIM, there exists a random element $X^C$ that can be sampled from a function $f : \mathcal{C} \times [0,1] \to \mathcal{X}$. Moreover, $T$ follows a uniform distribution $U[0,1]$ : i.e., $X^C = f(C, T)$, such that*

$$p_{Y|X^C}^{d_i}(\mathbf{y}|\mathbf{x}) = p_{Y|X^C}^{d_j}(\mathbf{y}|\mathbf{x}).$$

*Proof.* Given the FICIM or PICIM, we obtain the Markov chain $X - C - Y$. According to [Lemma A.1]() and [Proposition 1](), there exists a random element $X^C$ that can be sampled from a function $f : \mathcal{C} \times [0,1] \to \mathcal{X}$, such that

$$p_{X^C|C}^{d_i}(\mathbf{x}|\mathbf{c}) = p_{X^C|C}^{d_j}(\mathbf{x}|\mathbf{c}). \tag{1}$$

Now, we can rewrite the conditional probability $p_{Y|X^C}^{d_i}(\mathbf{y}|\mathbf{x})$ as follows:

$$p_{Y|X^C}^{d_i}(\mathbf{y}|\mathbf{x}) = \frac{\int_{\mathcal{C}} p_{Y|C}^{d_i}(\mathbf{y}|\mathbf{c}) p_{X^C|C}^{d_i}(\mathbf{x}|\mathbf{c}) d\mathbf{c}}{\int_{\mathcal{C}} p_{X^C|C}^{d_i}(\mathbf{x}|\mathbf{c}) d\mathbf{c}} \tag{2}$$

$$= \frac{\int_{\mathcal{C}} p_{Y|C}^{d_i}(\mathbf{y}|\mathbf{c}) p_{X^C|C}^{d_j}(\mathbf{x}|\mathbf{c}) d\mathbf{c}}{\int_{\mathcal{C}} p_{X^C|C}^{d_j}(\mathbf{x}|\mathbf{c}) d\mathbf{c}} \tag{3}$$

$$= p_{Y|X^C}^{d_j}(\mathbf{y}|\mathbf{x}), \tag{4}$$

where [Eq. (3)]() is derived from [Eq. (1)]() and [Eq. (4)]() follows by noting that the denominators are equal due to [Eq. (1)]().

$$\square$$

**Theorem A.2.** *([Theorem 2]()) Given a single source domain $S \in D$ and a target domain $T \in D$, and further assuming the label space $\mathcal{Y}_S = \mathcal{Y}_T = \mathcal{Y}$, we obtain a theoretical bound of the semantic controlled risk $R_T(Y|X^C)$, where $X^C$ can be sampled from a function $f : \mathcal{C} \times [0,1] \to \mathcal{X}_T$ as follows:*

$$R_T(Y|X^C) \leq (1 + \beta) R_S(Y|X^C),$$

*where $\beta = \sup_{C \in \mathcal{C}} p_C^T(\mathbf{c})/p_C^S(\mathbf{c}) - 1$.*

*Proof.*

$$R_T(Y|X^C)$$

$$= \iint\limits_{\mathcal{X},\mathcal{Y}} \ell(f(\mathbf{x}),\mathbf{y})p_{X^C}^T(\mathbf{x})p_{Y|X^C}^T(\mathbf{y}|\mathbf{x})d\mathbf{y}d\mathbf{x} \tag{5}$$

$$= \iint\limits_{\mathcal{X},\mathcal{Y}} \ell(f(\mathbf{x}),\mathbf{y})p_{X^C}^T(\mathbf{x})p_{Y|X^C}^T(\mathbf{y}|\mathbf{x})d\mathbf{y}d\mathbf{x} \tag{6}$$

$$- \iint\limits_{\mathcal{X},\mathcal{Y}} \ell(f(\mathbf{x}),\mathbf{y})p_{X^C}^T(\mathbf{x})p_{Y|X^C}^S(\mathbf{y}|\mathbf{x})d\mathbf{y}d\mathbf{x}$$

$$+ \iint\limits_{\mathcal{X},\mathcal{Y}} \ell(f(\mathbf{x}),\mathbf{y})p_{X^C}^T(\mathbf{x})p_{Y|X^C}^S(\mathbf{y}|\mathbf{x})d\mathbf{y}d\mathbf{x}$$

$$- \iint\limits_{\mathcal{X},\mathcal{Y}} \ell(f(\mathbf{x}),\mathbf{y})p_{X^C}^S(\mathbf{x})p_{Y|X^C}^S(\mathbf{y}|\mathbf{x})d\mathbf{y}d\mathbf{x}$$

$$+ \iint\limits_{\mathcal{X},\mathcal{Y}} \ell(f(\mathbf{x}),\mathbf{y})p_{X^C}^S(\mathbf{x})p_{Y|X^C}^S(\mathbf{y}|\mathbf{x})d\mathbf{y}d\mathbf{x}$$

$$= R_S(Y|X^C) \tag{7}$$

$$+ \iint\limits_{\mathcal{X},\mathcal{Y}} \ell(f(\mathbf{x}),\mathbf{y})p_{X^C}^S(\mathbf{x})p_{Y|X^C}^S(\mathbf{y}|\mathbf{x})(\frac{p_{X^C}^T(\mathbf{x})}{p_{X^C}^S(\mathbf{x})} - 1)d\mathbf{y}d\mathbf{x}$$

$$+ \iint\limits_{\mathcal{X},\mathcal{Y}} \ell(f(\mathbf{x}),\mathbf{y})p_{X^C}^T(\mathbf{x})(p_{Y|X^C}^T(\mathbf{y}|\mathbf{x}) - p_{Y|X^C}^S(\mathbf{y}|\mathbf{x}))d\mathbf{y}d\mathbf{x}$$

$$= R_S(Y|X^C) \tag{8}$$

$$+ \iint\limits_{\mathcal{X},\mathcal{Y}} \ell(f(\mathbf{x}),\mathbf{y})p_{X^C}^S(\mathbf{x})p_{Y|X^C}^S(\mathbf{y}|\mathbf{x})(\frac{p_{X^C}^T(\mathbf{x})}{p_{X^C}^S(\mathbf{x})} - 1)d\mathbf{y}d\mathbf{x}$$

$$\leq (1+\beta)R_S(Y|X^C), \tag{9}$$

where $\beta = \sup_{X^C \in \mathcal{X}} p_{X^C}^T(\mathbf{x})/p_{X^C}^S(\mathbf{x}) - 1 = \sup_{C \in \mathcal{C}} p_C^T(\mathbf{c})/p_C^S(\mathbf{c}) - 1$ from Lemma A.1. Eq. (5) follows the definition of the expected risk. We can obtain Eq. (6) by adding and subtracting two construction terms $\iint\limits_{\mathcal{X},\mathcal{Y}} \ell(f(\mathbf{x}),\mathbf{y})p_{X^C}^T(\mathbf{x})p_{Y|X^C}^S(\mathbf{y}|\mathbf{x})d\mathbf{y}d\mathbf{x}$ and $\iint\limits_{\mathcal{X},\mathcal{Y}} \ell(f(\mathbf{x}),\mathbf{y})p_{X^C}^S(\mathbf{x})p_{Y|X^C}^S(\mathbf{y}|\mathbf{x})d\mathbf{y}d\mathbf{x}$, respectively. The first part of Eq. (7) is the definition of the final part of Eq. (6). We obtain the second part of Eq. (7) by combining the third and fourth parts of Eq. (6). The final term of Eq. (7) is the combination of the first and second parts of Eq. (6). Eq. (8) can be obtained by $p_{Y|X^C}^T(\mathbf{y}|\mathbf{x}) = p_{Y|X^C}^S(\mathbf{y}|\mathbf{x})$ from Theorem A.1. By setting the $\sup_{X^C \in \mathcal{X}} p_{X^C}^T(\mathbf{x})/p_{X^C}^S(\mathbf{x})$ in the second part of Eq. (7) to be upper bounded by a constant $\beta + 1$, we obtain the inequality Eq. (9). $\qquad\square$

**Theorem A.3.** *(Theorem 3) Given a single source domain $S \in D$ and a target domain $T \in D$, and further assuming the label space $\mathcal{Y}_S \subset \mathcal{Y}_T$ and setting $\mathbf{y}^{uk}$ to represent the unknown target classes $\mathcal{Y}_T \setminus \mathcal{Y}_S$, we obtain a theoretical bound of the stable expected risk $R_T(Y|X^C)$ under OSDA, as follows:*

$$R_T(Y|X^C) \leq \underbrace{(1+\beta)R_S(Y|X^C)}_{\text{(1) Risk of known target classes}} + \underbrace{\int_{\mathcal{X}} \ell(f(\mathbf{x}),\mathbf{y}^{uk})p_{X^C Y}^T(\mathbf{x},\mathbf{y}^{uk})d\mathbf{x}}_{\text{(1) Risk of unknown target classes}},$$

*where $\beta = \sup_{C \in \mathcal{C}_S} p_C^T(\mathbf{c})/p_C^S(\mathbf{c}) - 1$.*

*Proof.*

$$R_T(Y|X^C)$$

$$= \iint_{\mathcal{X},\mathcal{Y}^S} \ell(f(\mathbf{x}),\mathbf{y})p_{X^C}^T(\mathbf{x})p_{Y|X^C}^T(\mathbf{y}|\mathbf{x})d\mathbf{y}d\mathbf{x} \tag{10}$$

$$+ \iint_{\mathcal{X},\mathcal{Y}^{uk}} \ell(f(\mathbf{x}),\mathbf{y})p_{X^C}^T(\mathbf{x})p_{Y|X^C}^T(\mathbf{y}|\mathbf{x})d\mathbf{y}d\mathbf{x}$$

$$= \iint_{\mathcal{X},\mathcal{Y}^S} \ell(f(\mathbf{x}),\mathbf{y})p_{X^C}^T(\mathbf{x})p_{Y|X^C}^T(\mathbf{y}|\mathbf{x})d\mathbf{y}d\mathbf{x}$$

$$+ \int_{\mathcal{X}} \ell(f(\mathbf{x}),\mathbf{y}^{uk})p_{X^CY}^T(\mathbf{x},\mathbf{y}^{uk})d\mathbf{x}$$

$$\leq (1+\beta)R_S(Y|X^C) \tag{11}$$

$$+ \int_{\mathcal{X}} \ell(f(\mathbf{x}),\mathbf{y}^{uk})p_{X^CY}^T(\mathbf{x},\mathbf{y}^{uk})d\mathbf{x},$$

where Eq. (10) is obtained by setting $\mathbf{y}^{uk}$ to represent $\mathcal{Y}_T \setminus \mathcal{Y}_S$, and Eq. (11) follows from Theorem A.2 such that $\beta = \sup_{C \in \mathcal{C}_S} p_C^T(\mathbf{c})/p_C^S(\mathbf{c}) - 1$. □

**Proposition A.1.** *Given a domain $d_i \in D$ and $d_i$ following the FICIM, there exists a random element $X^C$ that can be sampled from a function $f : \mathcal{C} \times [0,1] \to \mathcal{X}_{d_i}$ and T follows a uniform distribution $U[0,1]$, i.e., $X^C = f(C,T)$, such that*

$$R_{d_i}^*(Y|X) = R_{d_i}^*(Y|X^C).$$

*Proof.* The Markov chain for $d_i$ can be reduced to $X_{d_i} - C - Y_{d_i}$. There exists $X_{d_i}^C$ sampled from a function $f : \mathcal{C} \times [0,1] \to \mathcal{X}_{d_i}$. At this point, using Lemma A.1, we obtain

$$\mathbf{P}_{X^C|C}^{d_i} = \mathbf{P}_{X|C}^{d_i}. \tag{12}$$

From the FICIM property illustrated in Proposition 2, we obtain

$$\mathbf{P}_{X|C}^{d_i} = \mathbf{P}_{X|CY}^{d_i} \tag{13}$$

$$\mathbf{P}_{X^C|C}^{d_i} = \mathbf{P}_{X^C|CY}^{d_i}. \tag{14}$$

From Eq. (12), Eq. (13), and Eq. (14), we obtain

$$\mathbf{P}_{X|CY}^{d_i} = \mathbf{P}_{X^C|CY}^{d_i}, \tag{15}$$

which directly derives

$$\mathbf{P}_{XCY}^{d_i} = \mathbf{P}_{X^CCY}^{d_i}. \tag{16}$$

By integrating both sides of Eq. (16) with respect to $\mathcal{C}$, we obtain

$$\mathbf{P}_{XY}^{d_i} = \mathbf{P}_{X^CY}^{d_i}. \tag{17}$$

Thus, we can confirm that

$$R_{d_i}^*(Y|X) = \inf_{f \in \mathcal{F}, \varphi \in \Phi} \mathbf{E}_{\mathbf{P}_{XY}^{d_i}} \ell(f(\varphi(\mathbf{x})),\mathbf{y}) \tag{18}$$

$$= \inf_{f \in \mathcal{F}, \varphi \in \Phi} \mathbf{E}_{\mathbf{P}_{X^CY}^{d_i}} \ell(f(\varphi(\mathbf{x})),\mathbf{y}) \tag{19}$$

$$= R_{d_i}^*(Y|X^C), \tag{20}$$

where Eq. (18) and Eq. (20) follow from Definition 5, and Eq. (19) follows from Eq. (17). □

**Proposition A.2.** *Given an arbitrary domain $d_i \in D$ and $d_i$ following the PICIM, if we construct a new space $\mathcal{CV}$ by directly connecting space $\mathcal{C}$ and $\mathcal{V}$, there exists a random element $X^{CV}$ that can be sampled from a function $f : \mathcal{CV} \times [0,1] \to \mathcal{X}_{di}$ and $T$ follows uniform distribution $U[0,1]$, i.e., $X^C = f(CV, T)$, such that*

$$R^*_{d_i}(Y|X) = R^*_{d_i}(Y|X^{CV}).$$

*Proof.* The key difference between the FICIM and PICIM is that we do not have $\mathbf{P}^{d_i}_{X|CY} = \mathbf{P}^{d_i}_{X^C|CY}$ for the PICIM. However, if we construct a new element $CV$ by combining $C$ and $V$, we can obtain the causal structure $X - CV - Y$. Subsequently, we obtain

$$\mathbf{P}^{d_i}_{X|CV} = \mathbf{P}^{d_i}_{X|CVY}. \tag{21}$$

Using this equation, similar to the proof of [Proposition A.1](#), we obtain

$$R^*_{d_i}(Y|X) = R^*_{d_i}(Y|X^{CV}).$$

$\square$

**Lemma A.2.** *([Remark 1](#)) For [Theorem A.3](#), the second term of the bound $\int_{\mathcal{X}} \ell(f(\mathbf{x}), \mathbf{y}^{uk}) p^T_{X^C Y}(\mathbf{x}, \mathbf{y}^{uk}) d\mathbf{x}$ can be minimized by the optimal representation $Z^*_S = \varphi^*_S(X)$ that is obtained by ERM of a source domain $S$:*

$$\inf_{f \in \mathcal{F}} \int_{\mathcal{X}} \ell(f(\mathbf{x}), \mathbf{y}^{uk}) p^T_{X^C Y}(\mathbf{x}, \mathbf{y}^{uk}) d\mathbf{x}$$

$$= \inf_{g \in \mathcal{G}} \int_{\mathcal{X}} \ell(g(\varphi^*_S(\mathbf{x})), \mathbf{y}^{uk}) p^T_{X^C Y}(\mathbf{x}, \mathbf{y}^{uk}) d\mathbf{x}$$

*with the assumption that*

$$\mathcal{F} = \mathcal{G} \circ \Phi \qquad \forall f \in \mathcal{F}, g \in \mathcal{G}, \varphi \in \Phi.$$

*Proof.* The optimal feature classifier of the target domain based on the optimal source domain feature extractor is set as

$$g^*_T = \arg\inf_{g \in \mathcal{G}} \int_{\mathcal{X}} \ell(g(\varphi^*_S(\mathbf{x})), \mathbf{y}^{uk}) p^T_{X^C Y}(\mathbf{x}, \mathbf{y}^{uk}) d\mathbf{x}.$$

We obtain

$$\inf_{f \in \mathcal{F}} \int_{\mathcal{X}} \ell(f(\mathbf{x}), \mathbf{y}^{uk}) p^T_{X^C Y}(\mathbf{x}, \mathbf{y}^{uk}) d\mathbf{x}$$

$$\leq \int_{\mathcal{X}} \ell(g^*_T(\varphi^*_S(\mathbf{x})), \mathbf{y}^{uk}) p^T_{X^C Y}(\mathbf{x}, \mathbf{y}^{uk}) d\mathbf{x} \tag{22}$$

$$= \inf_{g \in \mathcal{G}} \int_{\mathcal{X}} \ell(g(\varphi^*_S(\mathbf{x})), \mathbf{y}^{uk}) p^T_{X^C Y}(\mathbf{x}, \mathbf{y}^{uk}) d\mathbf{x}. \tag{23}$$

[Eq. (22)](#) is obtained by the definition of infimum and [Eq. (23)](#) is determined by our setting of $g^*_T$. The optimal classifier that separates the unknown and known labels from $X^C$ is set as

$$f^*_T = (g_T \circ \varphi_T)^*$$

$$= \arg\inf_{g \in \mathcal{G}, \varphi \in \Phi} \int_{\mathcal{X}} \ell(g(\varphi(\mathbf{x})), \mathbf{y}^{uk}) p^T_{X^C Y}(\mathbf{x}, \mathbf{y}^{uk}) d\mathbf{x}.$$

Let $\tau : \mathcal{Z}_S^* \to \mathcal{X}_T^C$ can be any one-to-one function that exists in a weak condition $|\mathcal{X}_T^C| \geq |\mathcal{Z}_S^*| \geq 2$. As a result,

$$\inf_{f \in \mathcal{F}} \int_{\mathcal{X}} \ell(f(\mathbf{x}), \mathbf{y}^{uk}) p_{X^C Y}^T(\mathbf{x}, \mathbf{y}^{uk}) d\mathbf{x}$$

$$= \inf_{g \in \mathcal{G}, \varphi \in \Phi} \int_{\mathcal{X}} \ell(g(\varphi(\mathbf{x})), \mathbf{y}^{uk}) p_{X^C Y}^T(\mathbf{x}, \mathbf{y}^{uk}) d\mathbf{x} \tag{24}$$

$$= \int_{\mathcal{X}} \ell(f_T^*(\mathbf{x}), \mathbf{y}^{uk}) p_{X^C Y}^T(\mathbf{x}, \mathbf{y}^{uk}) d\mathbf{x} \tag{25}$$

$$= \int_{\mathcal{X}} \ell(f_T^*(\tau(\varphi_S^*(\mathbf{x}))), \mathbf{y}^{uk}) p_{X^C Y}^T(\mathbf{x}, \mathbf{y}^{uk}) d\mathbf{x} \tag{26}$$

$$\geq \int_{\mathcal{X}} \ell(g_T^*(\varphi_S^*(\mathbf{x})), \mathbf{y}^{uk}) p_{X^C Y}^T(\mathbf{x}, \mathbf{y}^{uk}) d\mathbf{x} \tag{27}$$

$$= \inf_{g \in \mathcal{G}} \int_{\mathcal{X}} \ell(g(\varphi_S^*(\mathbf{x}).\mathbf{y}^{uk}) p_{X^C Y}^T(\mathbf{x}, \mathbf{y}^{uk}) d\mathbf{x} \tag{28}$$

Eq. (24) is obtained by the assumption $\mathcal{F} = \mathcal{G} \circ \Phi$. Eq. (25) is determined by the setting of the optimal $f_T^*$. Eq. (26) is obtained from the existence of $\tau$. Eq. (27) and Eq. (28) follow from the definition of $g_T^*$ and infimum.

The combination of the opposite direction inequalities from Eq. (23) and Eq. (28) leads to the equality conclusion of this lemma. □

**Theorem A.4.** *(Theorem 4) For Theorem A.3, conducting the ERM on a FICIM source domain provides enough information to bound the stable expected risk of target domain $R_T(Y|X^C)$.*

*Proof.* From Proposition A.1, we obtain

$$R_{d_i}^*(Y|X) = R_{d_i}^*(Y|X^C)$$

for $d_i \in D$, and $d_i$ follows the FICIM.

As a source domain $S \in D$ and $S$ follows the FICIM, we obtain

$$R_S^*(Y|X) = R_S^*(Y|X^C).$$

Subsequently, for Theorem A.3, the first term $R_S(Y|X^C)$ can be minimized by ERM of a FICIM source domain $S$. Owing to Lemma A.2, the second term $\int_{\mathcal{X}} \ell(f(\mathbf{x}), \mathbf{y}^{uk}) p_{X^C Y}^T(\mathbf{x}, \mathbf{y}^{uk}) d\mathbf{x}$ can be minimized by the optimal representation $Z_S^* = \varphi_S^*(X)$ that is obtained by ERM of a source domain $S$.

Hence, the ERM of a FICIM source domain provides sufficient information to conduct subsequent searching for an optimal open-set classifier that provides adequate information to bound the stable expected risk of target domain $R_T(Y|X^C)$. □

**Theorem A.5.** *(Theorem 5) For Theorem A.3, conducting the ERM on a PICIM source domain can not bound the stable expected risk of target domain $R_T(Y|X^C)$.*

*Proof.* From Proposition A.2, we obtain

$$R_{d_i}^*(Y|X) = R_{d_i}^*(Y|X^{CV})$$

for $d_i \in D$ and $d_i$ follows the PICIM.

As a source domain $S \in D$ and $S$ follows the PICIM, we obtain

$$R_S^*(Y|X) = R_S^*(Y|X^{CV}).$$

Without further assumptions, we do not obtain $R_S^*(Y|X^C) = R_S^*(Y|X^{CV})$. Thus, we do not have $R_S^*(Y|X^C) = R_S^*(Y|X)$. Therefore, for Theorem A.3, the first term $R_S(Y|X^C)$ cannot be minimized by ERM of a PICIM source domain $S$. Hence, the ERM of a PICIM source domain cannot bound the stable expected risk of target domain $R_T(Y|X^C)$. □

**Theorem A.6.** *(Remark 2) Given a target domain $T \in D$, we determine a useful decomposition of the minimum expected risk as follows:*

$$R_T^*(Y|X) = \underbrace{R_T^*(Y|X^C)}_{\textit{(1) Minimum semantic controlled risk}} - \underbrace{[R_T^*(Y|X^C) - R_T^*(Y|X^{CV})]}_{\textit{(2) Uncontrolable spurious benefit}}. \tag{29}$$

*Proof.* We assume w.l.o.g that domain $T$ follows the PICIM. The results under this assumption can easily be generalized to the situation of the FICIM by setting $CV = C$. From $T \in D$ and Proposition A.2, we obtain

$$R_T^*(Y|X) = R_T^*(Y|X^{CV}). \tag{30}$$

We directly prove this theorem by adding and subtracting a term $R_T^*(Y|X^C)$. □

# D  EXPERIMENTAL DETAILS

This section provides further details about the three datasets and implementation details. All experimental results are the averages and variances obtained after running the experiments five times. The implementation is built upon the code open-sourced by Chen et al. (2021); Feder et al. (2023).

## D.1  OS-CMNIST DATA

### D.1.1  DATASET

We constructed our open-set CMNIST (OS-CMNIST) dataset following the dealing method of CMNIST (Arjovsky et al., 2020) to satisfy the setting demand of the OSDA task. We selected MNIST as our experimental dataset as it is an ideally clear dataset that does not include other attributes to determine the labels apart from the grayscale digits. A critical step in the creation of the OS-CMNIST training (source) domain was the random sampling of known and unknown classes from MNIST. If the data of the known classes were processed using the same method as CMNIST, we could obtain the PICIM source domain.

**Comparison groups.**   We designed the following common steps to construct the OS-CMNIST dataset: first, randomly sample $K$ known classes and $10 - K$ unknown classes from MNIST; second, assign a causal shape code $C$ to the known classes based on the digit: $C = 0$ for a random half of the $K$ classes and $C = 1$ for the other half; third, perform the same operation for unknown classes as in the second step.

Based on the known classes of the OS-CMNIST dataset, we constructed two comparison groups of the source domain, as follows:

- **FICIM source domain**: First, sample the variation color attribute code $V$ by flipping $C$ with a probability $1 - Corr(V, C)$; second, color the image green if $V = 1$ or red if $V = 0$.
- **PICIM source domain**: First, obtain the final class label $Y$ by flipping $C$ with a probability $1 - Corr(Y, C)$; second, sample the color attribute code $V$ by flipping $Y$ with a probability $1 - Corr(V, Y)$; finally, color the image green if $V = 1$ or red if $V = 0$.

In this case, $Corr(\cdot, \cdot)$ represents the correlation between two variables. As the binary label of a variable is changed by flipping another variable, the correlation relationship is the same for $Corr(\cdot, \cdot) < 0.5$ and $Corr(\cdot, \cdot) > 0.5$. It is known that these two variables are independent when $Corr(\cdot, \cdot) = 0.5$. When $Corr(\cdot, \cdot) > 0.5$, a greater $Corr(\cdot, \cdot)$ indicates a stronger correlation. When $Corr(\cdot, \cdot) < 0.5$, a smaller $Corr(\cdot, \cdot)$ indicates a stronger correlation. If $Corr(\cdot, \cdot) = 1$ or $= 0$, the two variables are perfectly positively correlated or negatively correlated. Note that the only difference between the FICIM and PICIM in our construction is that $Corr(Y, C) = 1$ for the FICIM, whereas $Corr(Y, C) < 1$ for the PICIM, which match the causal structure effectively. To ensure the comparability of the two groups further, we set a constant $\kappa_S = Corr(V, C)_S = Corr(V, Y)_S$ to represent the ratio of information acquired by color code $V$ for the source domain.

### D.1.2 EXPERIMENT SETTINGS

The aim of our main experiment was to demonstrate the influence of the ERM of a FICIM or PICIM source domain for the stable expected risk of the target domain.

**Structure of target domain.** As we aimed to examine the influence of the stable expected risk, the performance of the target domain should be most strongly related to the stable expected risk. According to Remark 2, the minimum risk $R_T^*(Y|X)$ is equal to the minimum stable expected risk $R_T^*(Y|X^C)$ for a FICIM target domain. Thus, we set the causal model of the target domain as the FICIM model. To obtain a FICIM target domain, we constructed two groups of six known classes, similar to the FICIM source domain, and added a group of four unknown classes with random colors.

**Loss function.** Similar to existing research (Chen et al., 2021), we adopted various loss functions, including ARPLoss, ARPLoss cs, RPLoss, GCPL, and Softmax loss function.

**Network structure.** We used the ResNet network architecture with 34 layers, and the state-of-the-art OSR algorithm from (Chen et al., 2021) to validate our theoretical results. We set the training parameters as follows: 40 epochs with a batch size of 64, the Momentum SGD optimizer, and a learning rate starting from 0.1 and decreasing by a factor of 0.1 every 30 epochs in the training process.

**Parameters of domain adaptation.** We set $Corr(Y, C)_S = 1$ for the FICIM source domain and $Corr(Y, C)_S = 0.75$ for the PICIM source domain to satisfy the properties of the FICIM and PICIM. Moreover, we set $\kappa_S = 0.8$ and $Corr(V, C)_T = 0.1$ to create distribution shifts between the source and target domains.

**Metrics.** Similar to (Dhamija et al., 2018; Chen et al., 2021), we combined three metrics to measure the classification performance in the target domain: closed-set accuracy (CS-ACC), area under the ROC curve (AUROC), and open-set classification rate (OSCR). For CS-ACC, AUROC, and OSCR, the larger value indicates better performance.

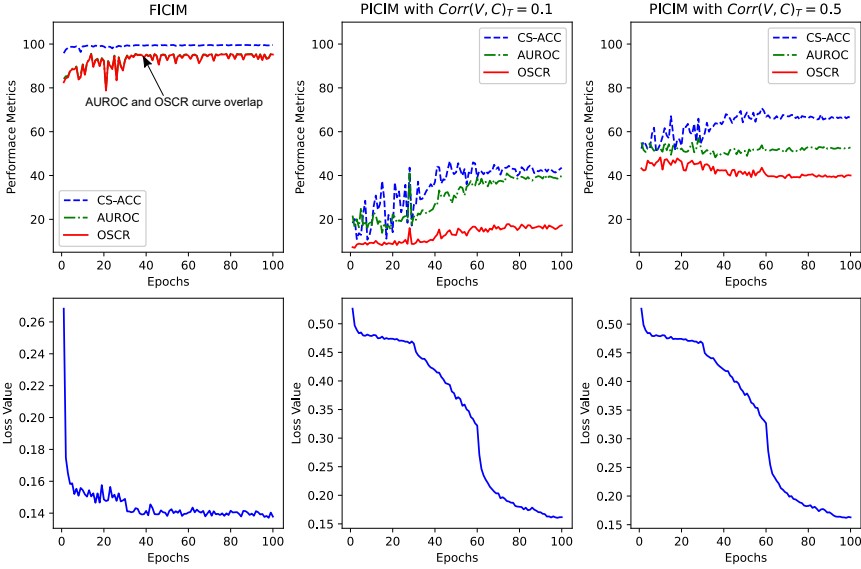

Figure 5: Performance metrics and loss values for FICIM and PICIM source domain. Left: FICIM source domain on target domain with $Corr(V, C)_T = 0.1$. Medium: PICIM source domain on target domain with $Corr(V, C)_T = 0.1$. Right: PICIM source domain on target domain with $Corr(V, C)_T = 0.5$.

### D.1.3  MAIN RESULTS

We plotted the curves of the epochs versus the performance metrics and loss value (Fig. 5) to verify the effectiveness of ERM on the FICIM and PICIM. It can be observed from the bottom three panels of Fig. 5 that all of these loss values converged over the epochs, but the FICIM loss converged faster. The top left panel of Fig. 5 indicates that the CS-ACC was near perfect for the ERM on the FICIM source domain, and the OSCR was entirely dependent on the AUROC. Moreover, the OSCR and AUROC were above $90\%$ most of the time, which demonstrates that the ERM of the FICIM could bound the stable expected risk of the OSDA. A comparison of the top three panels indicates that training on the PICIM source domain always performed worse than that on the FICIM, which supports the belief that the ERM of the PICIM could not bound the stable expected risk of the OSDA. An interesting observation for the PICIM source domain from the middle and right panels of Fig. 5 is that a stronger correlation between the shape $C$ and color $V$ for the target domain resulted in a stronger relationship between the CS-ACC and AUROC. When the $V$ and $C$ were independent, that is, $Corr(V, C) = 0.5$, the CS-ACC and AUROC were nearly independent. This observation challenges the opinion that the closed-set and open-set performances are highly correlated (Vaze et al., 2022). In contrast, the results from Fig. 5 demonstrate that the closed-set and open-set performance were highly correlated in two scenarios: when the source domain was the FICIM source domain and when both conditions were satisfied simultaneously; that is, the source domain was the PICIM source domain, and the target domain exhibited strong correlations between the variation and causal attributes.

It can be observed from Fig. 5 that the performance was almost stable from the 40th epoch. Thus, we compared the cross-sectional performance data of the FICIM and PICIM source domains at the 40th epoch. As indicated in Table 2, training with ERM on the FICIM source domain could achieve nearly perfect performance for the CS-ACC, AUROC, and OSCR of the target domain, which supports Theorem 4. We demonstrate Theorem 5 by observing that the CS-ACC, AUROC, and OSCR declined sharply from the FICIM source domain to the PICIM source domain. That is, training with ERM on the PICIM source domain resulted in a model that performed worse on the target domain than on the FICIM source domain.

**Additional results.**  To validate the effectiveness of our theoretical results in OOD tasks, we considered two different OOD scenarios. As shown in Table 6 and Table 7, there are significant performance differences between the models in the FICIM and PICIM scenarios, particularly in the TNR metric. This further confirms that the ERM of the FICIM could bound the stable expected risk of the OSDA.

Table 6: Distinguishing in- and out-of-distribution test set data for image classification under various validation setups. The known label categories are [6, 3, 4, 2, 8, 9], while the unknown label categories are [5, 0, 7, 1].

| Method | | TNR | AUROC | DTACC | AUIN | AUOUT |
|---|---|---|---|---|---|---|
| ARPLoss | FICIM | 81.18±0.19 | 95.46±0.19 | 90.04±0.19 | 96.43±0.19 | 93.03±0.19 |
| | PICIM | 3.13±0.19 | 52.22±1.19 | 56.88 ±0.08 | 66.63±0.67 | 39.17±0.88 |
| ARPLoss+CS | FICIM | 84.44±1.02 | 95.93±0.23 | 91.35±0.32 | 96.45±0.31 | 93.70 ±0.47 |
| | PICIM | 4.44±0.46 | 53.07±0.70 | 54.40±0.33 | 64.75±0.67 | 40.97± 0.50 |
| RPLOSS | FICIM | 75.94±5.67 | 91.97±3.76 | 87.82±3.80 | 92.67±3.50 | 90.13± 3.41 |
| | PICIM | 2.71±1.63 | 53.76±1.75 | 60.25±0.92 | 69.04±0.51 | 39.36±2.20 |
| Softmax | FICIM | 80.64±5.27 | 95.09±1.45 | 89.62±2.35 | 96.01±1.04 | 92.80±1.72 |
| | PICIM | 3.47±0.59 | 53.05±1.09 | 56.89±1.01 | 66.68±0.51 | 39.75 ±1.00 |
| GCPL | FICIM | 77.44±5.06 | 92.35±2.72 | 87.68±3.17 | 93.11±2.02 | 90.40±2.42 |
| | PICIM | 2.94±0.91 | 53.57±1.64 | 58.58±0.67 | 67.95±0.69 | 39.56±1.27 |

The results of five randomized trials with 40 epochs were averaged. For the FICIM group, the key parameters were $\kappa = 0.8, Corr(V, C)_T = 0.1$. For the PICIM group, the key parameters were $\kappa = 0.8, Corr(Y, C)_S = 0.75, Corr(V, C)_T = 0.1$.

Table 7: Distinguishing in- and out-of-distribution test set data for image classification under various validation setups. The known label categories are [3, 7, 4, 0, 8, 5], while the unknown label categories are [2, 6, 9, 1].

| Method | | TNR | AUROC | DTACC | AUIN | AUOUT |
|---|---|---|---|---|---|---|
| ARPLoss | FICIM | 88.16±0.79 | 97.34±0.05 | 92.20±0.31 | 98.15±0.06 | 95.77±0.29 |
| | PICIM | 3.13±0.66 | 52.22±2.73 | 56.88±2.24 | 66.63±3.05 | 39.17±1.67 |
| ARPLoss+CS | FICIM | 89.41±1.06 | 97.66±0.18 | 92.90±0.43 | 98.38±0.15 | 96.25±0.26 |
| | PICIM | 4.82±0.46 | 51.47±1.79 | 52.33±1.19 | 61.18±1.63 | 41.73±1.12 |
| RPLOSS | FICIM | 90.93±0.37 | 97.67±0.27 | 93.39±0.38 | 98.11±0.41 | 96.71 ±0.27 |
| | PICIM | 2.25±0.70 | 52.41±2.43 | 59.24±1.81 | 65.61±1.65 | 39.33±1.63 |
| Softmax | FICIM | 87.41±0.81 | 97.30±0.11 | 92.09±0.27 | 98.11±0.05 | 95.88±0.26 |
| | PICIM | 3.72±0.21 | 50.01±2.73 | 53.72±2.79 | 60.40±3.92 | 39.80±1.05 |
| GCPL | FICIM | 91.32±0.55 | 96.74±0.13 | 93.49±0.43 | 96.61±0.28 | 95.88±0.15 |
| | PICIM | 2.58±0.55 | 51.70±2.94 | 57.32±1.77 | 64.96±1.55 | 39.42±1.96 |

### D.1.4 SENSITIVITY ANALYSES

This section presents extensive additional experiments. By modifying several key control variables, such as the number of unknown classes, $\kappa_S$, $Corr(V, C)_T$, and loss functions, while keeping other variables consistent with the main experiment, we tested the sensitivity of our theoretical results to uncertainties under different conditions.

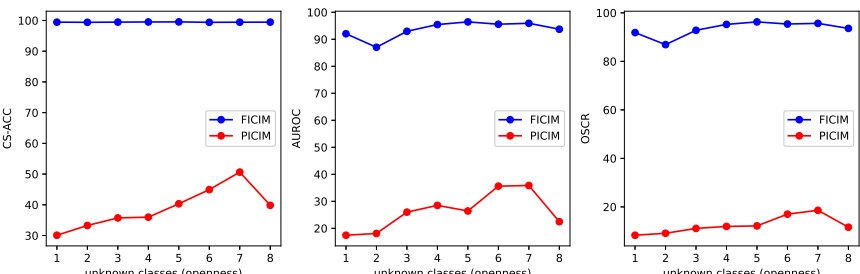

Figure 6: Performance variation trend comparison between FICIM and PICIM with number of unknown classes (openness).

The first important sensitivity parameter to consider was the number of unknown classes, which is a proxy variable of openness. Scheirer et al. (2013) first introduced the concept of openness for the OSR problem. For a fixed number of testing classes, increasing the number of unknown classes in the training stage increases the openness. Hence, we used the number of unknown classes belonging to $\{1, \ldots, 8\}$ as the proxy variable of openness. Note that the FICIM and PICIM settings were the same as those in Table 2 except for the unknown classes. The openness results of Fig. 6 were obtained from models that were trained for 40 epochs. Fig. 6 indicates that the FICIM source domain tended to perform much better than the PICIM source domain for all numbers of unknown classes, and hence, changing the openness did not change our theoretical results. Interestingly, this figure indicates that all of the performance metrics remained steady as the unknown classes increased for the FICIM source domain, whereas no clear trend was observed for the PICIM source domain. This finding reveals that it is unnecessary to overthink openness for the open-set task.

The other critical parameters were as follows: $\kappa_S \in [0, 1]$, $Corr(V, C)_T \in [0, 1]$, loss functions (Softmax/ARPLoss (Chen et al., 2021)/GCPLoss (Yang et al., 2020)/ARPLoss+CS (Chen et al., 2021)). Note that ARPLoss+CS is not a pure loss function, but ARPLoss with confusing samples. For simplicity, we added ARPLoss+CS to the group of loss functions. For these two parameters with an interval of $[0, 1]$, we selected the parameters belonging to $\{0.0, 0.2, 0.4, 0.5, 0.6, 0.8, 1\}$ to conduct the experiments. It can be observed from Fig. 7a, 7b, and 7c that the PICIM source domain

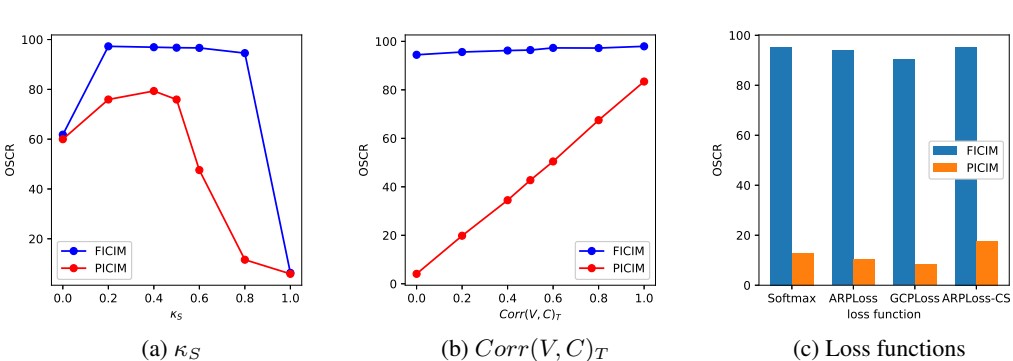

(a) $\kappa_S$        (b) $Corr(V, C)_T$        (c) Loss functions

Figure 7: Performance variation trend comparison of FICIM and PICIM

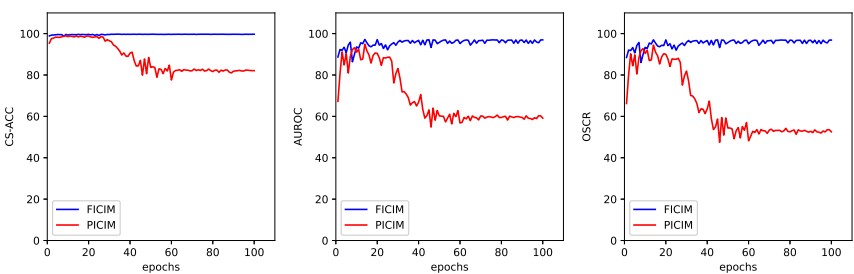

Figure 8: Performance variation trend comparison of between FICIM and PICIM on OS-BMNIST.

always performed worse than the FICIM source domain. This indicates that none of these parameters affected the results of the main experiment.

Moreover, several significant findings emerged from the experimental results. First, Fig. 7a shows that the FICIM exhibited the same performance as the PICIM when $\kappa_S = 0$ or $1$; that is, the color attribute $V$ was perfectly positively correlated or negatively correlated with the label $Y$. This observation may be because ERM cannot distinguish the causal $C$ and variation $V$ when $V$ is perfectly associated with $Y$ in the source domain. Second, according to Fig. 7b, with the increase in $Corr(V, C)_T$ of the target domain, the performance of the PICIM also increased, while the performance of the FICIM remained stable. This result demonstrates that the ERM on the PICIM source domain learned the variation information, which could play a more important role if $Corr(V, C)_T$ increased. Moreover, this finding verifies that the ERM on the FICIM source domain learned the causally invariant information owing to its performance independence with $Corr(V, C)_T$. Third, Fig. 7c indicates that softmax performed the best if ARPLoss-CS was not considered. Surprisingly, ARPLoss-CS achieved the best performance for the PICIM source domain, while achieving nearly the same performance as softmax for the FICIM source domain. This finding confirms that the generation of confusing training samples can provide additional information to aid in training.

Finally, we considered another dataset, namely the open-set binary MNIST (OS-BMNIST), to enhance the generality of our results. This dataset is similar to OS-CMNIST, but without color, which means that it does not contain spurious attributes. Hence, the PICIM variation factors only contained noise by setting $Corr(Y, C) < 1$. In this case, w.l.o.g., we set $Corr(Y, C) = 0.75$ for the PICIM. As illustrated in Fig. 8, following convergence of the metrics, the overall performance of the FICIM was much better than that of the PICIM source domain, which supports Theorem 4 and Theorem 5. In contrast to the PICIM patterns in Fig. 5, the performance of the PICIM for OS-BMNIST increased to a high level in the first several epochs and subsequently maintained this high level but finally decreased sharply to a stable low level. A possible explanation for this is that although full training of ERM can result in performance crashes owing to noise variations, light training of ERM can provide sufficient information to bound the general error. This concept has been applied in many methods,

Table 8: Performance comparison of FICIM and PICIM source domain on Synthetic Data.

| Method | | AUC | ACC | F1 |
|---|---|---|---|---|
| ERM | FICIM | $97.01 \pm 0.33$ | $90.85 \pm 0.54$ | $90.85 \pm 0.55$ |
| | PICIM | $76.23 \pm 6.42$ | $73.10 \pm 4.86$ | $72.83 \pm 5.14$ |
| Reweighting | FICIM | $97.01 \pm 0.33$ | $90.85 \pm 0.54$ | $90.85 \pm 0.55$ |
| | PICIM | $77.73 \pm 5.87$ | $78.25 \pm 4.65$ | $78.21 \pm 4.45$ |
| Aug($\lambda = 0$) | FICIM | $96.95 \pm 0.37$ | $90.74 \pm 0.61$ | $90.74 \pm 0.63$ |
| | PICIM | $96.91 \pm 0.45$ | $90.61 \pm 0.80$ | $90.60 \pm 0.84$ |
| Aug($\lambda = 0.2$) | FICIM | $96.96 \pm 0.35$ | $90.77 \pm 0.59$ | $90.77 \pm 0.60$ |
| | PICIM | $92.23 \pm 1.43$ | $83.47 \pm 1.98$ | $83.43 \pm 2.11$ |
| Aug($\lambda = 0.3$) | FICIM | $96.98 \pm 0.36$ | $90.80 \pm 0.60$ | $90.80 \pm 0.61$ |
| | PICIM | $94.17 \pm 0.86$ | $86.18 \pm 1.29$ | $86.14 \pm 1.41$ |
| Aug($\lambda = 0.4$) | FICIM | $97.00 \pm 0.36$ | $90.82 \pm 0.60$ | $90.82 \pm 0.61$ |
| | PICIM | $96.12 \pm 0.40$ | $89.26 \pm 0.61$ | $89.24 \pm 0.69$ |
| Aug($\lambda = 0.5$) | FICIM | $97.00 \pm 0.36$ | $90.82 \pm 0.62$ | $90.82 \pm 0.63$ |
| | PICIM | $96.83 \pm 0.42$ | $90.45 \pm 0.68$ | $90.43 \pm 0.74$ |

such as regularization and early stopping. Although the spurious attributes and noise all belonged to the variation attributes $V$, a further comparison of Fig. 5 and Fig. 8 reveals that the spurious attributes had a much more serious impact on the target domain performance than the noise attributes.

### D.2 SYNTHETIC DATA

#### D.2.1 DATASET

To further validate the effectiveness of our theoretical framework, we conducted experiments on synthetic data. Following the experimental setups of existing studies (Feder et al., 2023), we generate synthetic data for a binary classification problem where $|V| = 8$ (cardinality of V). We sample $P(V|Y)$ to simulate varying degrees of spurious correlations. Then we draw $x = [x^*, x_{\text{spu}}]$ from a Gaussian distribution,

$$x_i = \begin{bmatrix} x^* \\ x_{\text{spu},i} \end{bmatrix} \backsim \mathcal{N} \left( \begin{bmatrix} \mu_{y_i} \\ \mu_{c_i} \end{bmatrix}, \begin{bmatrix} \sigma^2 \mathbf{I_{d^*}} & 0 \\ 0 & \sigma^2_{\text{spu}} \mathbf{I_{d_c}} \end{bmatrix} \right) .$$

In our simulations, we set core dimension $d^* = 10$, spurious feature dimension $d_{\text{spu}} = 300$ and $\sigma^2_{\text{spu}} = 0.05$, $\sigma = d^*$ to make the maxmargin classifiers depend on the spurious features. The parameters $\mu_{y_i}$, $\mu_{c_i}$ are drawn uniformly from a sphere of norm 1/3 and 60, respectively. For the corruptions of augmentations where we add $\xi_i(\mu_c - \mu_{c_i})$, the $\xi_i$ variables are drawn from a truncated Gaussian centered at $\lambda$ with standard deviation 0.1.

#### D.2.2 EXPERIMENT SETUP

In the experiments, we calculate the mutual information between two categorical variables based on the joint probability table. We use logistic regression to fit the model under all conditions, employing the Adam optimizer.

#### D.2.3 ADDITIONAL RESULTS

To further validate our theoretical results, we selected the model performance under different conditions when the mutual information $I(Y; V)$ is 0.62. As shown in Table 8, traditional ERM and reweighting methods are significantly affected by different data generation mechanisms. Even with advanced augmentation models, training with ERM on the PICIM source domain resulted in a model that performed worse on the target domain than on the FICIM source domain.

## D.3 RESTAURANT REVIEW DATA

### D.3.1 DATASET

We use the CEBaB dataset (Abraham et al., 2022), which consists of short restaurant reviews and ratings from OpenTable, including evaluations for food, service, noise, ambiance, and an overall rating. For our experiments, we used the train-exclusive split of the dataset, which contains 1, 755 examples. To analyze the data, we transformed the overall rating into a binary outcome. The original rating scale ranges from 1 to 5, and we classified a rating of 3 or higher as 1, and anything below as 0. We utilized a bag-of-words model with CountVectorizer and fitted logistic regression models from the sklearn library.

### D.3.2 EXPERIMENT SETUP

Following the counterfactual generation procedure in (Feder et al., 2023), we generate counterfactual restaurant reviews conditional on food rating and overall rating. For each review, we first find a set of matched examples. We then select the subset that has different food-mention attribute and prompt GPT-4 to rewrite. This results in 956 augmentations. Counterfactual enhancement should capture what the review would look like if the reviewer were more concise or less concise. Following existing research (Feder et al., 2023), we generate counterfactual restaurant reviews conditional on food and overall ratings. We find matched examples for each review, select those with different food-mentions, and prompt a GPT-4 to rewrite them, reflecting how the reviews would appear if the reviewer was more/less concise. The template for generating counterfactual prompts for restaurant reviews is shown in Figure D.3.2.

To further validate the effectiveness of our theoretical results, we conducted fine-tuning of large models based on restaurant reviews. For our experiments, we used the train-inclusive split of the dataset, which contains 11,728 examples. Similar to the processing workflow for food-mentions in restaurant reviews, we performed matching based on rating-noise and rating-overall, and then utilized GPT-4 for rewriting the restaurant reviews. The original restaurant review data satisfies the PICIM, while the generated counterfactual data satisfies the FICIM. We fine-tuned three large models using different sample sizes $n = \{1000, 2000, 3000, 4000, 5000\}$. The fine-tuning instructions for the templates are shown in Figure D.3.2.

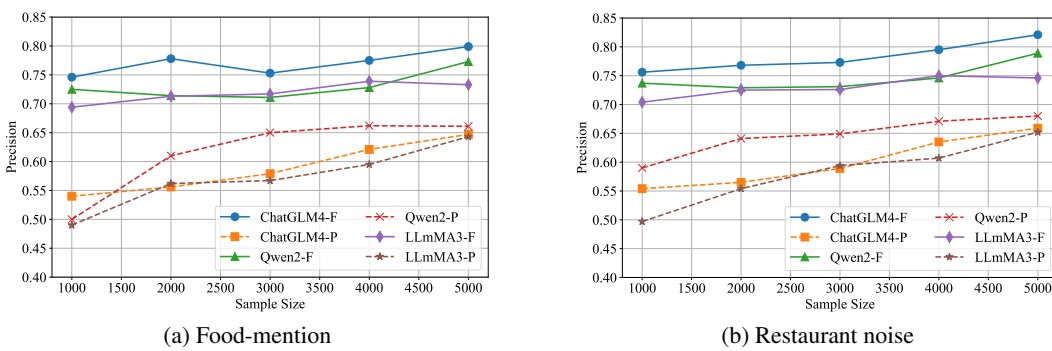

(a) Food-mention        (b) Restaurant noise

Figure 9: Performance comparison of fine-tuning.

**Prompt**

**Input:** `""" You are a very helpful, diligent, and intelligent language model assistant. Your task is to generate counterfactual versions of restaurant reviews, specifically how the review would change if specific food items were mentioned or omitted. You will be given an original restaurant review and a comparator review. You only need to rewrite the food section of the original review. If the comparator review mentions specific food items, ensure the rewritten review includes the same items; if the original review mentions specific food items but the comparator does not, remove them from the rewritten version. The overall rating should align with the comparator review, considering ambiance, food, noise, and service.`

`--- EXAMPLE INPUT - START ---`

`original_review: [Original_review],`
`original_ratings: [score: Score]`

`compare_reviews: [Original_review1],`
`compare_ratings: [score: Score1]`

`--- EXAMPLE INPUT - END ---`

`"""`

---

**Output:**
```
{
original_review: __,
rewrite_score: __,
rewrite_review: __
}
```

**Fine-tuning instruction pairs**

**Instruction**

`"You are a very helpful, diligent, and intelligent language model assistant. Your task is to rate restaurants based on their reviews, with scores of either 0 or 1. The rating primarily considers four aspects: ambiance, food, noise, and service."`

**Input**

"The steak is very fresh and delicious; the restaurant is quiet with a great atmosphere."

**Output**

1

### D.3.3 Additional results

The experimental results of fine-tuning based on food mentions and restaurant reviews are shown in Fig. 4 and Fig. 9. We can draw the following two main conclusions: (1) training on the FICIM source domain always perform better than that on the PICIM, which supports the belief that the ERM of the PICIM could not bound the stable expected risk of the OSDA, while the FICIM can; (2) utilizing our proposed FICIM causal model, high-quality data can be filtered to facilitate the efficient pre-training and fine-tuning of large models.

