# OpenReview forum: "A Causal Theoretical Framework for Open Set Domain Adaptation"
_ICLR.cc/2025/Conference — Submitted to ICLR 2025_

### Official Review · Reviewer_vJud · 2024-11-02

**Soundness:** 1
**Presentation:** 2
**Contribution:** 2
**Rating:** 3
**Confidence:** 4

**Summary:**

The paper studies open set domain adaptation from a causal view. Two frameworks, Fully Informative Causal Invariance Model (FICIM)and the Partially Informative Causal Invariance Model (PICIM), are proposed. Some theoretical bounds are obtained, accompanied by experimental results.

**Strengths:**

An interesting and meaningful problem, with both theoretical and empirical resutls.

**Weaknesses:**

The assumption is TOO strong or unrealistic, making the paper meaningless.

**Questions:**

This review is going to be short, as the paper is a clear reject in my view.

My major concern is that the assumption is TOO strong or unrealistic. In Assumption 1, invariant attributes are required to satisfy that both $P(Y|C)$ and $P(X|C)$ are invariant across domains. Notice that, $X$ is the input data, e.g., an image. I never see this assumption in any related work before, and such an assumption is also unrealistic. Let $X$ be the image of dogs sitting on different backgrounds  in different domains, and the primary object of interest is the dog. In this case, the semantic feature of dog could be the invariant feature, but the distribution of the images changes across the domains as the background changes.  Also, in the two frameworks in Figure 2, both $C$ and $V$ are parents of $X$, so $P(X|C)$ in general should not be the same, as it also depends on $V$.



I was thinking if it was only a typo. However, while I just went over the rest of the paper, it seems that $P(X|C)$ being invariant is used multiple times in the theoretic derivations.

I may have a misunderstanding about this assumption.  If authors can provide examples or scenarios where this assumption hold in practice, I would like to re-evaluate the paper during rebuttal period.

---

### Official Review · Reviewer_9gCK · 2024-11-02

**Soundness:** 2
**Presentation:** 1
**Contribution:** 1
**Rating:** 3
**Confidence:** 3

**Summary:**

This paper attempts to categorize the Open Set Domain Adaptation (OSDA) problem into two models: Fully Informative Causal Invariance Model (FICIM) and Partially Informative Causal Invariance Model (PICIM). The authors claim that ERM performs well when trained on a source domain of the FICIM type, as the expected risk $R_T$ in the target domain is bounded by the expected risk $R_S(Y|X^C)$ in the source domain (Theorem 3). In contrast, ERM performs poorly on datasets of the PICIM type (Theorem 5). Furthermore, the authors validate their theoretical findings on CMNIST, synthetic datasets, and restaurant review (text) data.

**Strengths:**

Although the authors attempted to provide proofs for the proposed theorems and included a comprehensive appendix, I still find the notation system overly confusing (deviating from commonly used mathematical expressions in this field), creating significant reading difficulties.

**Weaknesses:**

I believe one of the important contributions of the paper is the introduction of the PICIM and FICIM concepts. This classification approach is already a common consensus in the field of invariant learning. However, the paper’s presentation is rather disorganized, and the clarity of expression is insufficient, making it difficult to read. Additionally, I feel that the authors have not developed a meaningful method for adapting OSDA based on their theoretical framework, making the contribution somewhat lacking. Any effort to introduce new ideas is certainly welcome, but unfortunately, I cannot recommend accepting this paper.

**Issues with PICIM and FICIM**:

- The concepts of PICIM and FICIM proposed in this paper seem to originate from the FIIF and PIIF models introduced by [Ahuja et al., 2021], which also focus on modeling the data generation process. This concept has been widely cited in the invariant learning literature, such as in [Liu et al., 2021] and [Chen et al., 2022]. What is the variable $Z$ in Figure 2? I searched the entire paper but could not find a definition for $Z$. Moreover, the bidirectional arrows between $Y$ and $V$ are not clearly defined, and the types of arrows are explicitly described in Section 2.6 of [Causality, 2009]. I suggest that the authors add the environment variable $E$ to the causal graph to represent changes in causal relationships across different domains.
- I am confused about Theorem 3. In the causal graph of FICIM, there still exists an association between $C$ and $V$. How does the ERM algorithm ensure that invariant features and spurious features can be successfully disentangled? In other words, we can only obtain the expected risk $R_S(Y|X)$, and ERM cannot achieve the stable expected risk $R_S(Y|X^C)$. This seems to contradict the results of [Ahuja et al., 2021] (Theorems 3 and 4).

[Ahuja et al., 2021]. "Invariance principle meets information bottleneck for out-of-distribution generalization." *Advances in Neural Information Processing Systems* 34 (2021): 3438-3450.

[Liu et al., 2021]. "Learning causal semantic representation for out-of-distribution prediction." *Advances in Neural Information Processing Systems* 34 (2021): 6155-6170.

[Chen et al., 2022]. "Learning causally invariant representations for out-of-distribution generalization on graphs." *Advances in Neural Information Processing Systems* 35 (2022): 22131-22148.

Pearl, Judea. *Causality*. Cambridge university press, 2009.

**Unclear Notation**:

- The notation system in the paper is extremely confusing, and using $P_X$ and $P_{X|Y}$ seems meaningless. For instance, in line 218,  $p_{Y|X^C}^{d_i}(y|x) = p^{d_j}_{Y|X^C}(y|x)$---are you trying to express that the domain of $x$ remains invariant as element $X^C$? Wouldn’t it be clearer to denote cross-domain invariance as $P^{d_i}(Y|X = X^C) = P^{d_j}(Y|X = X^C)$?
- Definitions 5-6 appear to be redundant, reiterating well-known concepts in the field, seemingly to pad the length of the paper.
- In lines 282-283, the subscript $C \in \mathcal{C}_S$ of $\sup$ is not displayed correctly as a subscript. Additionally, $p^TC(c)$ and $p^SC(c)$ are miswritten, and the same symbol error appears in line 300 with $p^TC(c)$.

**Other Issues**:

- I don’t quite understand how the "Risk of unknown target classes" in Theorem 3 is calculated. Under FIIF, ERM cannot disentangle $C$ and $V$, so how can you ensure that $Z^*_S$ originates from $X^C$?

**Questions:**

- In Proposition 1, regarding the definition of domain invariance, is $X$ truly domain-invariant given $C$? Referring to the causal graph in [Chen et al., 2022], introducing the domain variable $D$, which points to $V$, would imply that given $C$, the chain structure $X \leftarrow V \leftarrow D$ indicates that $X$ is related to $D$, i.e., $P^{d_i}(X|C) \neq P^{d_j}(X|C)$.
- In Definition 6, the expected risk is defined, but in line 217, the function $f$ is said to map to the invariant element $X^C$, whereas in the loss function $\ell(f(x),y)$, $f$ seems to map to the label. Are both of these functions denoted by the same symbol $f$?

---

### Official Review · Reviewer_Si1Z · 2024-11-04

**Soundness:** 2
**Presentation:** 2
**Contribution:** 2
**Rating:** 5
**Confidence:** 3

**Summary:**

This paper proposes two classes of causal models (FICIM and PICIM) for the data generating process in order to understand distribution shift in Open Set Domain Adaptation. Based on these causal models, they derive bounds on the risk on the target set given the risk on the source set using ERM. They theoretically demonstrate that the target risk for FICIM is bounded, however for PICIM it may be unbounded. Empirical experiments verify these claims.

**Strengths:**

1)The problem is important and relevant to Open Set Domain Adaptation.
2)The proposed method and results are novel.
3)The experimental results are strong verify the theoretical results.

**Weaknesses:**

1)P(X∣C) need not be invariant. For example, consider the causal diagram of FICIM. Since there is a bidirectional arrow between C and V, it is not necessary for P(X∣C) to remain invariant between the source and target domains.

(Theorems 2 and 4) The constant β could be quite large in many cases. While these theorems establish that the risk is bounded, they do not fully explain why ERM performs so well, nor why other domain adaptation algorithms might not achieve better performance.

**Questions:**

See Weaknesses.

---

### Meta-Review · Area_Chair_fWfa · 2024-12-20

**Metareview:**

This paper introduces two causal models, FICIM and PICIM, to study distribution shifts in Open Set Domain Adaptation (OSDA) and provides theoretical bounds on target risk using ERM. While the problem is relevant, reviewers raised concerns about the practicality of assumptions, overlap with prior work, and insufficient novelty or meaningful application to OSDA. Additionally, unclear presentation and confusing notation detract from the paper’s impact.

**Additional Comments On Reviewer Discussion:**

The authors did not provide any rebuttals.

---

### Decision · Program_Chairs · 2025-01-22

Reject